# Hyperparameter Ensembles for Robustness and Uncertainty Quantification

**Florian Wenzel, Jasper Snoek, Dustin Tran, Rodolphe Jenatton**
Google Research
{florianwenzel, jsnoek, trandustin, rjenatton}@google.com

## Abstract

Ensembles over neural network weights trained from different random initialization, known as deep ensembles, achieve state-of-the-art accuracy and calibration. The recently introduced batch ensembles provide a drop-in replacement that is more parameter efficient. In this paper, we design ensembles not only over weights, but over hyperparameters to improve the state of the art in both settings. For best performance independent of budget, we propose *hyper-deep ensembles*, a simple procedure that involves a random search over different hyperparameters, themselves stratified across multiple random initializations. Its strong performance highlights the benefit of combining models with both weight *and* hyperparameter diversity. We further propose a parameter efficient version, *hyper-batch ensembles*, which builds on the layer structure of batch ensembles and self-tuning networks. The computational and memory costs of our method are notably lower than typical ensembles. On image classification tasks, with MLP, LeNet, ResNet 20 and Wide ResNet 28-10 architectures, we improve upon both deep and batch ensembles.

## 1 Introduction

Neural networks are well-suited to form *ensembles* of models [30]. Indeed, neural networks trained from different random initialization can lead to equally well-performing models that are nonetheless *diverse* in that they make complementary errors on held-out data [30]. This property is explained by the multi-modal nature of their loss landscape [24] and the randomness induced by both their initialization and the stochastic methods commonly used to train them [8, 38, 9].

Many mechanisms have been proposed to further foster diversity in ensembles of neural networks, e.g., based on cyclical learning rates [36] or Bayesian analysis [17]. In this paper, we focus on exploiting the diversity induced by combining neural networks defined by different hyperparameters. This concept is already well-established [13] and the auto-ML community actively applies it [21, 65, 53, 46]. We build upon this research with the following two complementary goals.

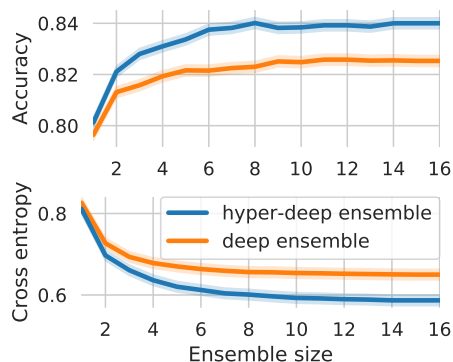

Figure 1: Comparison of our hyper-deep ensemble with deep ensemble for different ensemble sizes using a Wide ResNet 28-10 over CIFAR-100. Combining models with different hyperparameters is beneficial.

First, for performance independent of computational and memory budget, we seek to improve upon *deep ensembles* [43], the current state-of-the-art ensembling method in terms of robustness and uncertainty quantification [64, 28]. To this end, we develop a simple stratification scheme which combines random search and the greedy selection of hyperparameters from [13] with the benefit

of multiple random initializations per hyperparameter like in deep ensembles. Figure 1 illustrates our algorithm for a Wide ResNet 28-10 where it leads to substantial improvements, highlighting the benefits of combining different initialization *and* hyperparameters.

Second, we seek to improve upon *batch ensembles* [69], the current state-of-the-art in efficient ensembles. To this end, we propose a parameterization combining that of [69] and self-tuning networks [52], which enables both weight and hyperparameter diversity. Our approach is a drop-in replacement that outperforms batch ensembles and does not need a separate tuning of the hyperparameters.

## 1.1 Related work

**Ensembles over neural network weights.** Combining the outputs of several neural networks to improve their single performance has a long history, e.g., [47, 30, 25, 41, 58, 15]. Since the quality of an ensemble hinges on the diversity of its members [30], many mechanisms were developed to generate diverse ensemble members. For instance, cyclical learning-rate schedules can explore several local minima [36, 76] where ensemble members can be snapshot. Other examples are MC dropout [23] or the random initialization itself, possibly combined with the bootstrap [45, 43]. More generally, Bayesian neural networks can be seen as ensembles with members being weighted by the (approximated) posterior distribution over the parameters [34, 51, 56, 7, 71, 72].

**Hyperparameter ensembles.** Hyperparameter-tuning methods [20] typically produce a pool of models from which ensembles can be constructed post hoc, e.g., [65]. This idea has been made systematic as part of `auto-sklearn` [21] and successfully exploited in several other contexts, e.g., [19] and specifically for neural networks [53] as well as in computer vision [60] and genetics [35]. In particular, the greedy ensemble construction from [13] (and later variations thereof [12]) was shown to work best among other algorithms, either more expensive or more prone to overfitting. To the best of our knowledge, such ensembles based on hyperparameters have not been studied in the light of predictive uncertainty. Moreover, we are not aware of existing methods to efficiently build such ensembles, similarly to what batch ensembles do for deep ensembles. Finally, recent research in Bayesian optimization has also focused on directly optimizing the performance of the ensemble while tuning the hyperparameters [46].

Hyperparameter ensembles also connect closely to probabilistic models over structures. These works often analyze Bayesian nonparametric distributions, such as over depth and width of a neural network, leveraging Markov chain Monte Carlo for inference [37, 1, 18, 42]. In this work, we examine more parametric assumptions, building on the success of variational inference and mixture distributions: for example, the validation step in hyper-batch ensemble can be viewed as a mixture variational posterior and the entropy penalty is the ELBO's KL divergence toward a uniform prior.

Concurrent to our paper, [75] construct neural network ensembles within the context of neural architecture search, showing improved robustness for predictions with distributional shift. One of their methods, NES-RS, has similarities with our hyper-deep ensembles (see Section 3), also relying on both random search and [13] to form ensembles, but do not stratify over different initializations. We vary the hyperparameters while keeping the architecture fixed while [75] study the converse. Furthermore, [75] do not explore a parameter- and computationally-efficient method (see Section 4).

**Efficient hyperparameter tuning & best-response function.** Some hyperparameters of a neural network, e.g., its $L_2$ regularization parameter(s), can be optimized by estimating the *best-response* function [26], i.e., the mapping from the hyperparameters to the parameters of the neural networks solving the problem at hand [11]. Learning this mapping is an instance of learning an hypernetwork [61, 62, 29] and falls within the scope of bilevel optimization problems [14]. Because of the daunting complexity of this mapping, [50, 52] proposed scalable local approximations of the best-response function. Similar methodology was also employed for style transfer and image compression [3, 16]. The *self-tuning networks* from [52] are an important building block of our approach wherein we extend their setting to the case of an ensemble over different hyperparameters.

## 1.2 Contributions

We examine two regimes to exploit hyperparameter diversity: **(a)** ensemble performance independent of budget and **(b)** ensemble performance seeking parameter *efficiency*, where, respectively, deep and batch ensembles [43, 69] are state-of-the-art. We propose one ensemble method for each regime:

**(a) Hyper-deep ensembles.** We define a greedy algorithm to form ensembles of neural networks exploiting two sources of diversity: varied hyperparameters and random initialization. By stratifying models with respect to the latter, our algorithm subsumes deep ensembles that we outperform in our experiments. Our approach is a simple, strong baseline that we hope will be used in future research.

**(b) Hyper-batch ensembles.** We efficiently construct ensembles of neural networks defined over different hyperparameters. Both the ensemble members *and* their hyperparameters are learned end-to-end in a single training procedure, directly maximizing the ensemble performance. Our approach outperforms batch ensembles and generalizes the layer structure of [52] and [69], while keeping their original memory compactness and efficient minibatching for parallel training and prediction.

We illustrate the benefits of our two ensemble methods on image classification tasks, with multi-layer perceptron, LeNet, ResNet 20 and Wide ResNet 28-10 architectures, in terms of both predictive performance and uncertainty. The code for generic hyper-batch ensemble layers can be found in `https://github.com/google/edward2` and the code to reproduce the experiments of Section 5.2 is part of `https://github.com/google/uncertainty-baselines`.

## 2 Background

We introduce notation and background required to define our approach. Consider an i.i.d. classification setting with data $\mathcal{D} = \{(\mathbf{x}_n, y_n)\}_{n=1}^N$ where $\mathbf{x}_n \in \mathbb{R}^d$ is the feature vector corresponding to the $n$-th example and $y_n$ its class label. We seek to learn a classifier in the form of a neural network $f_{\boldsymbol{\theta}}$ where all its parameters (weights and bias terms) are summarized in $\boldsymbol{\theta} \in \mathbb{R}^p$. In addition to its primary parameters $\boldsymbol{\theta}$, the model $f_{\boldsymbol{\theta}}$ will also depend on $m$ hyperparameters that we refer to as $\boldsymbol{\lambda} \in \mathbb{R}^m$. For instance, an entry in $\boldsymbol{\lambda}$ could correspond to the dropout rate of a given layer in $f_{\boldsymbol{\theta}}$.

Equipped with some loss function $\ell$, e.g., the cross entropy, and some regularization term $\Omega(\cdot, \boldsymbol{\lambda})$, e.g., the squared $L_2$ norm with a strength defined by an entry of $\boldsymbol{\lambda}$, we are interested in

$$\hat{\boldsymbol{\theta}}(\boldsymbol{\lambda}) \in \underset{\boldsymbol{\theta} \in \mathbb{R}^p}{\arg\min} \ \mathbb{E}_{(\mathbf{x},y) \in \mathcal{D}} \big[ \mathcal{L}(\mathbf{x}, y, \boldsymbol{\theta}, \boldsymbol{\lambda}) \big] \quad \text{with} \quad \mathcal{L}(\mathbf{x}, y, \boldsymbol{\theta}, \boldsymbol{\lambda}) = \ell(f_{\boldsymbol{\theta}}(\mathbf{x}, \boldsymbol{\lambda}), y) + \Omega(\boldsymbol{\theta}, \boldsymbol{\lambda}), \quad (1)$$

where $\mathbb{E}_{(\mathbf{x},y) \in \mathcal{D}}[\cdot]$ stands for the expectation with a uniform distribution over $\mathcal{D}$. As we shall see in Section 5, the loss $\ell = \ell_{\boldsymbol{\lambda}}$ can also depend on $\boldsymbol{\lambda}$, for instance to control a label smoothing parameter [67]. In general, $\boldsymbol{\lambda}$ is chosen based on some held-out evaluation metric by grid search, random search [6] or more sophisticated hyperparameter-tuning methods [20].

### 2.1 Deep ensembles and batch ensembles

Deep ensembles [43] are a simple ensembling method where neural networks with different random initialization are combined. Deep ensembles lead to remarkable predictive performance and robust uncertainty estimates [64, 28]. Given some hyperparameters $\boldsymbol{\lambda}_0$, a deep ensemble of size $K$ amounts to solving $K$ times (1) with random initialization and aggregating the outputs of $\{f_{\hat{\boldsymbol{\theta}}_k(\boldsymbol{\lambda}_0)}(\cdot, \boldsymbol{\lambda}_0)\}_{k=1}^K$.

Batch ensembles [69] are a state-of-the-art *efficient* alternative to deep ensembles, preserving their performance while reducing their computational and memory burden. To simplify the presentation, we focus on the example of a dense layer in $f_{\boldsymbol{\theta}}$, with weight matrix $\mathbf{W} \in \mathbb{R}^{r \times s}$ where $r$ and $s$ denote the input and output dimensions of the layer respectively.

A deep ensemble of size $K$ needs to train, predict with, and store $K$ weight matrices $\{\mathbf{W}_k\}_{k=1}^K$. Instead, batch ensembles consider a *single* matrix $\mathbf{W} \in \mathbb{R}^{r \times s}$ together with two sets of auxiliary vectors $[\mathbf{r}_1, \ldots, \mathbf{r}_K] \in \mathbb{R}^{r \times K}$ and $[\mathbf{s}_1, \ldots, \mathbf{s}_K] \in \mathbb{R}^{s \times K}$ such that the role of $\mathbf{W}_k$ is played by

$$\mathbf{W} \circ (\mathbf{r}_k \mathbf{s}_k^\top) \ \text{ for each } \ k \in \{1, \ldots, K\}, \quad (2)$$

where we denote by $\circ$ the element-wise product (which we will broadcast row-wise or column-wise depending on the shapes at play). Not only does (2) lead to a memory saving, but it also allows for efficient minibatching, where each datapoint may use a different ensemble member. Given a batch of inputs $\mathbf{X} \in \mathbb{R}^{b \times r}$, the predictions for the $k$-th member equal $\mathbf{X}[\mathbf{W} \circ (\mathbf{r}_k \mathbf{s}_k^\top)] = [(\mathbf{X} \circ \mathbf{r}_k^\top)\mathbf{W}] \circ \mathbf{s}_k^\top$. By properly tiling the batch $\mathbf{X}$, the $K$ members can thus predict in parallel in one forward pass [69].

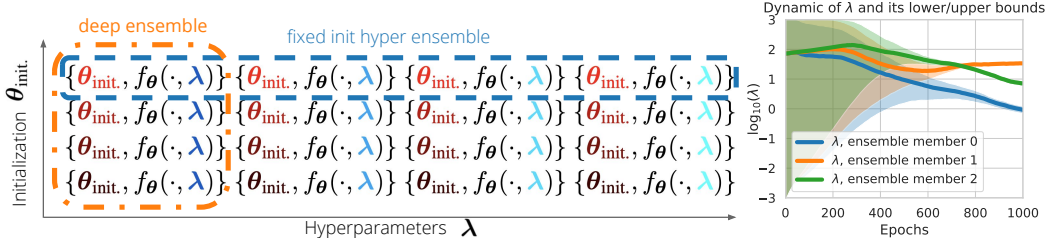

Figure 2: LEFT: Pictorial view of deep ensemble ("column") and fixed init hyper ensemble ("row") for models $f_{\boldsymbol{\theta}}(\cdot, \boldsymbol{\lambda})$ with parameters $\boldsymbol{\theta}$ and hyperparameters $\boldsymbol{\lambda}$. Our new method *hyper-deep ensemble* can search in the whole "block", exploiting both initialization and hyperparameter diversity. RIGHT: Example of the optimization path of *hyper-batch ensemble* for an entry of the hyperparameters $\boldsymbol{\lambda}$ (the $L_2$ parameter of an MLP over CIFAR-100) with its upper/lower bounds (shaded regions). The lower/upper bounds of the three members converge to a diverse set of hyperparameters.

## 2.2 Self-tuning networks

Hyperparameter tuning typically involves multiple runs of the training procedure. One efficient alternative [50, 52] is to approximate the best-response function, i.e., the mapping from $\boldsymbol{\lambda}$ to optimal parameters $\hat{\boldsymbol{\theta}}(\boldsymbol{\lambda})$. The local approximation of [52] captures the changes of $\boldsymbol{\lambda}$ by scaling and shifting the hidden units of $f_{\boldsymbol{\theta}}$, which requires in turn extra parameters $\boldsymbol{\theta}' \in \mathbb{R}^{p'}$, summarized in $\boldsymbol{\Theta} = \{\boldsymbol{\theta}, \boldsymbol{\theta}'\}$. [52] call the resulting approach *self-tuning network* since $f_{\boldsymbol{\Theta}}$ tunes online its own hyperparameters $\boldsymbol{\lambda}$. In the sequel, $\boldsymbol{\lambda}$ will be continuous such as dropout rates, $L_2$ penalties and label smoothing.

**Example of the dense layer.** We illustrate the choice and role of $\boldsymbol{\theta}'$ in the example of a dense layer (the convolutional layer is similar to [59]; see details in [52]). The weight matrix $\mathbf{W} \in \mathbb{R}^{r \times s}$ and bias $\mathbf{b} \in \mathbb{R}^s$ of a dense layer are defined as (with $\boldsymbol{\Delta}$ and $\boldsymbol{\delta}$ of the same shapes as $\mathbf{W}$ and $\mathbf{b}$ respectively),

$$\mathbf{W}(\boldsymbol{\lambda}) = \mathbf{W} + \boldsymbol{\Delta} \circ \mathbf{e}(\boldsymbol{\lambda})^{\top} \quad \text{and} \quad \mathbf{b}(\boldsymbol{\lambda}) = \mathbf{b} + \boldsymbol{\delta} \circ \mathbf{e}'(\boldsymbol{\lambda}), \tag{3}$$

where $\mathbf{e}(\boldsymbol{\lambda}) \in \mathbb{R}^s$ and $\mathbf{e}'(\boldsymbol{\lambda}) \in \mathbb{R}^s$ are real-valued embeddings of $\boldsymbol{\lambda}$. In [52], the embedding is linear, i.e., $\mathbf{e}(\boldsymbol{\lambda}) = \mathbf{C}\boldsymbol{\lambda}$ and $\mathbf{e}'(\boldsymbol{\lambda}) = \mathbf{C}'\boldsymbol{\lambda}$. In this example, we have original parameters $\boldsymbol{\theta} = \{\mathbf{W}, \mathbf{b}\}$ as well as the additional parameters $\boldsymbol{\theta}' = \{\boldsymbol{\Delta}, \boldsymbol{\delta}, \mathbf{C}, \mathbf{C}'\}$.

**Training objective.** Since $\boldsymbol{\theta}'$ captures changes in $\boldsymbol{\theta}$ induced by changes in $\boldsymbol{\lambda}$, [50, 52] replace the typical objective (1), defined for a *single* value of $\boldsymbol{\lambda}$, with an *expected* objective [50, 52, 16],

$$\min_{\boldsymbol{\Theta} \in \mathbb{R}^{p+p'}} \mathbb{E}_{\boldsymbol{\lambda} \sim p(\boldsymbol{\lambda}), (\mathbf{x}, y) \in \mathcal{D}} \big[ \mathcal{L}(\mathbf{x}, y, \boldsymbol{\Theta}, \boldsymbol{\lambda}) \big], \tag{4}$$

where $p(\boldsymbol{\lambda})$ denotes some distribution over the hyperparameters $\boldsymbol{\lambda}$. When $p$ is kept fixed during the optimization of (4), the authors of [50] observed that $\hat{\boldsymbol{\theta}}(\boldsymbol{\lambda})$ is not well approximated and proposed instead to use a distribution $p_t(\boldsymbol{\lambda}) = p(\boldsymbol{\lambda} | \boldsymbol{\xi}_t)$ varying with the iteration $t$. In our work we choose $p(\cdot | \boldsymbol{\xi}_t)$ to be a log-uniform distribution with $\boldsymbol{\xi}_t$ containing the bounds of the ranges of $\boldsymbol{\lambda}$ (see Section 4). The key benefit from (4) is that a *single* (though, more costly) training gives access to a mapping $\boldsymbol{\lambda} \mapsto f_{\hat{\boldsymbol{\Theta}}}(\cdot, \boldsymbol{\lambda})$ which approximates the behavior of $f_{\hat{\boldsymbol{\Theta}}}$ for hyperparameters in the support of $p(\boldsymbol{\lambda})$.

**Alternating optimization.** The procedure followed by [52] consists in alternating between training and tuning steps. First, the training step performs a stochastic gradient update of $\boldsymbol{\Theta}$ in (4), jointly sampling $\boldsymbol{\lambda} \sim p(\boldsymbol{\lambda} | \boldsymbol{\xi}_t)$ and $(\mathbf{x}, y) \in \mathcal{D}$. Second, the tuning step makes a stochastic gradient update of $\boldsymbol{\xi}_t$ by minimizing some *validation* objective (e.g., the cross entropy):

$$\min_{\boldsymbol{\xi}_t} \mathbb{E}_{\boldsymbol{\lambda} \sim p(\boldsymbol{\lambda} | \boldsymbol{\xi}_t), (\mathbf{x}, y) \in \mathcal{D}_{\text{val}}} \big[ \ell_{\text{val}}(f_{\boldsymbol{\Theta}}(\mathbf{x}, \boldsymbol{\lambda}), y) \big]. \tag{5}$$

In (5), derivatives are taken through samples $\boldsymbol{\lambda} \sim p(\boldsymbol{\lambda} | \boldsymbol{\xi}_t)$ by applying the reparametrization trick [39]. To prevent $p(\boldsymbol{\lambda} | \boldsymbol{\xi}_t)$ from collapsing to a degenerate distribution, and inspired by variational inference, the authors of [52] add an entropy regularization term $\mathcal{H}[\cdot]$ controlled by $\tau \geq 0$ so that (5) becomes

$$\min_{\boldsymbol{\xi}_t} \mathbb{E}_{\boldsymbol{\lambda} \sim p(\boldsymbol{\lambda} | \boldsymbol{\xi}_t), (\mathbf{x}, y) \in \mathcal{D}_{\text{val}}} \big[ \ell_{\text{val}}(f_{\boldsymbol{\Theta}}(\mathbf{x}, \boldsymbol{\lambda}), y) - \tau \mathcal{H}[p(\boldsymbol{\lambda} | \boldsymbol{\xi}_t)] \big]. \tag{6}$$

# 3 Hyper-deep ensembles

Figure 2-(left) visualizes different models $f_{\boldsymbol{\theta}}(\cdot, \boldsymbol{\lambda})$ according to their hyperparameters $\boldsymbol{\lambda}$ along the $x$-axis and their initialization $\boldsymbol{\theta}_{\text{init.}}$ on the $y$-axis. In this view, a deep ensemble corresponds to a "column" where models with different random initialization are combined together, for a fixed $\boldsymbol{\lambda}$. On the other hand, a "row" corresponds to the combination of models with different hyperparameters. Such a "row" typically stems from the application of some hyperparameter-tuning techniques [20].

**Fixed initialization hyper ensembles.** Given the simplicity, broad applicability, and performance of the greedy algorithm from [13]—e.g., in auto-ML settings [21], we use it as our canonical procedure to generate a "row", i.e., an ensemble of neural networks with fixed parameter initialization and various hyperparameters. We refer to it as *fixed init hyper ensemble*. For completeness, we recall the procedure from [13] in Appendix A (Algorithm 2, named hyper_ens). Given an input set of models (e.g., from random search), hyper_ens greedily grows an ensemble until some target size $K$ is met by selecting the model with the best improvement of some score, e.g., the validation log-likelihood. We select the models *with replacement* to be able to learn weighted combinations thereof (see Section 2.1 in [13]). Note that the procedure from [13] does not require the models to have a fixed initialization: we consider here a fixed initialization to isolate the effect of just varying the hyperparameters (while deep ensembles vary only the initialization, with fixed hyperparameters).

Our goal is two-fold: (a) we want to demonstrate the complementarity of random initialization and hyperparameters as sources of diversity in the ensemble, and (b) design a simple algorithmic scheme that exploits both sources of diversity while encompassing the construction of deep ensembles as a subcase. We defer to Section 5 the study of (a) and next focus on (b).

**Hyper-deep ensembles.** We proceed in three main steps, as summarized in Algorithm 1. In lines 1-2, we first generate one "row" according to hyper_ens based on the results of random search [6] as input. We then tile and stratify that "row" by training the models for different random initialization (see lines 4-7). The resulting set of models is illustrated in Figure 2-(left). In line 10, we finally re-apply hyper_ens on that stratified set of models to extract an ensemble that can exploit the two sources of diversity. By design, a deep ensemble is one possible outcome of this procedure—one "column"—and so is *fixed init hyper ensemble* described in the previous paragraph—one "row".

In lines 1-2, running random search leads to a set of $\kappa$ models (i.e., $\mathcal{M}_0$). If we were to stratify all of them, we would need $K$ seeds for each of those $\kappa$ models, hence a total of $\mathcal{O}(\kappa K)$ models to train. However, we first apply hyper_ens to extract $K$ models out of the $\kappa$ available ones, with $K \ll \kappa$. The stratification then needs $K$ seeds for each of those $K$ models (lines 4-7), thus $\mathcal{O}(K^2)$ models to train. We will see in Section 5 that even with standard hyperparameters, e.g., dropout or $L_2$ parameters, Algorithm 1 can lead to substantial improvements over deep ensembles. In Appendix C.7.5,

---
**Algorithm 1:** hyper_deep_ens$(K, \kappa)$

1  $\mathcal{M}_0 = \{f_{\boldsymbol{\theta}_j}(\cdot, \boldsymbol{\lambda}_j)\}_{j=1}^{\kappa} \leftarrow$ rand_search$(\kappa)$;
2  $\mathcal{E}_0 \leftarrow$ hyper_ens$(\mathcal{M}_0, K)$ and $\mathcal{E}_{\text{strat.}} = \{\ \}$;
3  **foreach** $f_{\boldsymbol{\theta}}(\cdot, \boldsymbol{\lambda}) \in \mathcal{E}_0$.unique() **do**
4      **foreach** $k \in \{1, \ldots, K\}$ **do**
5          $\boldsymbol{\theta}' \leftarrow$ random initialization;
6          $f_{\boldsymbol{\theta}_k}(\cdot, \boldsymbol{\lambda}) \leftarrow$ train $f_{\boldsymbol{\theta}'}(\cdot, \boldsymbol{\lambda})$;
7          $\mathcal{E}_{\text{strat.}} = \mathcal{E}_{\text{strat.}} \cup \{\ f_{\boldsymbol{\theta}_k}(\cdot, \boldsymbol{\lambda})\}$;
8      **end**
9  **end**
10 **return** hyper_ens$(\mathcal{E}_{\text{strat.}}, K)$;

---

we conduct ablation studies to relate to the top-$K$ strategy used in [60] and NES-RS from [75].

# 4 Hyper-batch ensembles

This section presents our efficient approach to construct ensembles over different hyperparameters.

## 4.1 Composing the layer structures of batch ensembles and self-tuning networks

The core idea lies in the *composition* of the layers used by batch ensembles [69] for ensembling parameters and self-tuning networks [52] for parameterizing the layer as an explicit function of hyperparameters. The composition preserves complementary features from both approaches.

We continue the example of the dense layer from Section 2.1-Section 2.2. The convolutional layer is described in Appendix B.1. Assuming an ensemble of size $K$, we have for $k \in \{1, \ldots, K\}$

$$\mathbf{W}_k(\boldsymbol{\lambda}_k) = \mathbf{W} \circ (\mathbf{r}_k \mathbf{s}_k^\top) + [\Delta \circ (\mathbf{u}_k \mathbf{v}_k^\top)] \circ \mathbf{e}(\boldsymbol{\lambda}_k)^\top \ \text{ and } \ \mathbf{b}_k(\boldsymbol{\lambda}_k) = \mathbf{b}_k + \boldsymbol{\delta}_k \circ \mathbf{e}'(\boldsymbol{\lambda}_k), \quad (7)$$

where the $\mathbf{r}_k$'s (respectively, $\mathbf{u}_k$'s) in $\mathbb{R}^r$ and $\mathbf{s}_k$'s (respectively, $\mathbf{v}_k$'s) in $\mathbb{R}^s$ are vectors which diversify the shared matrix $\mathbf{W}$ (respectively, $\mathbf{\Delta}$) in $\mathbb{R}^{r \times s}$; and the $\mathbf{b}_k$'s in $\mathbb{R}^s$ and $\boldsymbol{\delta}_k$'s in $\mathbb{R}^s$ are the bias terms for each of the $K$ ensemble members. We comment on some important properties of (7):

- As noted by [69], formulation (2) includes a set of rank-1 factors which diversify individual ensemble member weights. In (7), the rank-1 factors $\mathbf{r}_k\mathbf{s}_k^\top$ and $\mathbf{u}_k\mathbf{v}_k^\top$ capture this weight diversity for each respective term.

- As noted by [52], formulation (3) captures local hyperparameter variations in the vicinity of some $\boldsymbol{\lambda}$. The term $[\mathbf{\Delta} \circ (\mathbf{u}_k\mathbf{v}_k^\top)] \circ \mathbf{e}(\boldsymbol{\lambda}_k)^\top$ in (7) extends this behavior to the vicinity of the $K$ hyperparameters $\{\boldsymbol{\lambda}_1, \dots, \boldsymbol{\lambda}_K\}$ indexing the $K$ ensemble members.

- Equation (7) maintains the compactness of the original layers of [52, 69] with a resulting memory footprint about twice as large as [69] and equivalent to [52] up to the rank-1 factors.

- Given $K$ hyperparameters $\{\boldsymbol{\lambda}_1, \dots, \boldsymbol{\lambda}_K\}$ and a batch of inputs $\mathbf{X} \in \mathbb{R}^{b \times r}$, the structure of (7) preserves the efficient minibatching of [69]. If $\mathbf{1}_b$ is the vector of ones in $\mathbb{R}^b$, we can tile $\mathbf{X}$, $\mathbf{1}_b\boldsymbol{\lambda}_k^\top$ and $\mathbf{1}_b\mathbf{e}(\boldsymbol{\lambda}_k)^\top$, enabling all $K$ members to predict in a *single* forward pass.

- From an implementation perspective, (7) enables direct reuse of existing code, e.g., `DenseBatchEnsemble` and `Conv2DBatchEnsemble` from [68]. The implementation of our layers can be found in https://github.com/google/edward2.

## 4.2   Objective function: from single model to ensemble

We first need to slightly overload the notation from Section 2.2 and we write $f_{\boldsymbol{\Theta}}(\mathbf{x}, \boldsymbol{\lambda}_k)$ to denote the prediction for the input $\mathbf{x}$ of the $k$-th ensemble member indexed by $\boldsymbol{\lambda}_k$. In $\boldsymbol{\Theta}$, we pack all the parameters of $f$, as those described in the example of the dense layer in Section 4.1. In particular, predicting with $\boldsymbol{\lambda}_k$ is understood as using the corresponding parameters $\{\mathbf{W}_k(\boldsymbol{\lambda}_k), \mathbf{b}_k(\boldsymbol{\lambda}_k)\}$ in (7).

**Training and validation objectives.**   We want the ensemble members to account for a diverse combination of hyperparameters. As a result, each ensemble member is assigned its *own* distribution of hyperparameters, which we write $p_t(\boldsymbol{\lambda}_k) = p(\boldsymbol{\lambda}_k|\boldsymbol{\xi}_{k,t})$ for $k \in \{1, \dots, K\}$. Along the line of (4), we consider an expected training objective which now simultaneously operates over $\boldsymbol{\Lambda}_K = \{\boldsymbol{\lambda}_k\}_{k=1}^K$

$$\min_{\boldsymbol{\Theta}} \mathbb{E}_{\boldsymbol{\Lambda}_K \sim q_t, (\mathbf{x},y) \in \mathcal{D}} \Big[ \mathcal{L}(\mathbf{x}, y, \boldsymbol{\Theta}, \boldsymbol{\Lambda}_K) \Big] \ \text{ with } \ q_t(\boldsymbol{\Lambda}_K) = q(\boldsymbol{\Lambda}_K|\{\boldsymbol{\xi}_{k,t}\}_{k=1}^K) = \prod_{k=1}^K p_t(\boldsymbol{\lambda}_k) \quad (8)$$

and where $\mathcal{L}$, compared with (1), is extended to handle the ensemble predictions

$$\mathcal{L}(\mathbf{x}, y, \boldsymbol{\Theta}, \boldsymbol{\Lambda}_K) = \ell\big(\{f_{\boldsymbol{\Theta}}(\mathbf{x}, \boldsymbol{\lambda}_k)\}_{k=1}^K, y\big) + \Omega\big(\boldsymbol{\Theta}, \{\boldsymbol{\lambda}_k\}_{k=1}^K\big).$$

For example, the loss $\ell$ can be the ensemble cross entropy or the average ensemble-member cross entropy (in our experiments, we will use the latter as recent results suggests it often generalizes better [17]). The introduction of one distribution $p_t$ per ensemble member also affects the validation step of the alternating optimization, in particular we adapt (6) to become

$$\min_{\{\boldsymbol{\xi}_{k,t}\}_{k=1}^K} \mathbb{E}_{\boldsymbol{\Lambda}_K \sim q_t, (\mathbf{x},y) \in \mathcal{D}_{\text{val}}} \Big[ \ell_{\text{val}}(\{f_{\boldsymbol{\Theta}}(\mathbf{x}, \boldsymbol{\lambda}_k)\}_{k=1}^K, y) - \tau \mathcal{H}\big[q_t(\boldsymbol{\Lambda}_K)\big] \Big]. \quad (9)$$

Note that the extensions (8)-(9) with $K = 1$ fall back to the standard formulation of [52]. In our experiments, we take $\Omega$ to be $L_2$ regularizers applied to the parameters $\mathbf{W}_k(\boldsymbol{\lambda}_k)$ and $\mathbf{b}_k(\boldsymbol{\lambda}_k)$ of each ensemble member. In Appendix B.2, we show how to efficiently vectorize the computation of $\Omega$ across the ensemble members and mini-batches of $\{\boldsymbol{\lambda}_k\}_{k=1}^K$ sampled from $q_t$, as required by (8). In practice, we use one sample of $\boldsymbol{\Lambda}_K$ for each data point in the batch: for MLP/LeNet (Section 5.1), we use 256, while for ResNet-20/W. ResNet-28-10 (Section 5.2), we use 512 (64 for each of 8 workers).

**Definition of $p_t$.**   In the experiments of Section 5, we will manipulate hyperparameters $\boldsymbol{\lambda}$ that are positive and bounded (e.g., a dropout rate). For each ensemble member with hyperparameters $\boldsymbol{\lambda}_k \in \mathbb{R}^m$, we thus define its distribution $p_t(\boldsymbol{\lambda}_k) = p(\boldsymbol{\lambda}_k|\boldsymbol{\xi}_{k,t})$ to be $m$ independent *log-uniform distributions* (one per dimension in $\boldsymbol{\lambda}_k$), which is a standard choice for hyperparameter tuning, e.g., [5, 6, 53]. With this choice, $\boldsymbol{\xi}_{k,t}$ contains $2m$ parameters, namely the bounds of the ranges of

Table 1: Comparison over CIFAR-100 and Fashion MNIST with MLP and LeNet models. We report means ± standard errors (over the 3 random seeds and pooled over the 2 tuning settings). "single" stands for the best between `rand search` and `Bayes opt`. "fixed init ens" is a shorthand for `fixed init hyper ens`, i.e., a "row" in Figure 2-(left). We separately compare the *efficient* methods (3 rightmost columns) and we mark in bold the best results (within one standard error). Our two methods hyper-deep/hyper-batch ensembles improve upon deep/batch ensembles respectively (in Appendix C.7.2, we assess the statistical significance of those improvements with a Wilcoxon signed-rank test, paired along settings, datasets and model types).

| | | single (1) | fixed init ens (3) | hyper-deep ens (3) | deep ens (3) ‖ | batch ens (3) | STN (1) | hyper-batch ens (3) |
|---|---|---|---|---|---|---|---|---|
| cifar100 (mlp) | nll ↓ | 2.977 ± 0.010 | **2.943** ± 0.010 | **2.953** ± 0.058 | **2.969** ± 0.057 ‖ | 3.015 ± 0.003 | 3.029 ± 0.006 | **2.979** ± 0.004 |
| | acc ↑ | 0.277 ± 0.002 | **0.287** ± 0.003 | **0.291** ± 0.004 | **0.289** ± 0.003 ‖ | 0.275 ± 0.001 | 0.268 ± 0.002 | **0.281** ± 0.002 |
| | ece ↓ | **0.034** ± 0.008 | **0.029** ± 0.007 | **0.022** ± 0.007 | **0.038** ± 0.014 ‖ | **0.022** ± 0.002 | 0.033 ± 0.004 | 0.030 ± 0.002 |
| cifar100 (lenet) | nll ↓ | **2.399** ± 0.204 | **2.259** ± 0.067 | **2.211** ± 0.066 | **2.334** ± 0.141 ‖ | 2.350 ± 0.024 | 2.329 ± 0.017 | **2.283** ± 0.016 |
| | acc ↑ | 0.420 ± 0.011 | **0.439** ± 0.008 | **0.452** ± 0.007 | **0.421** ± 0.026 ‖ | **0.438** ± 0.003 | 0.415 ± 0.003 | 0.428 ± 0.003 |
| | ece ↓ | **0.064** ± 0.036 | **0.049** ± 0.023 | **0.039** ± 0.013 | **0.050** ± 0.015 ‖ | 0.058 ± 0.015 | **0.024** ± 0.007 | 0.058 ± 0.004 |
| fmnist (mlp) | nll ↓ | 0.323 ± 0.003 | **0.312** ± 0.003 | **0.310** ± 0.001 | 0.319 ± 0.005 ‖ | 0.351 ± 0.004 | 0.316 ± 0.003 | **0.308** ± 0.002 |
| | acc ↑ | 0.889 ± 0.002 | **0.893** ± 0.001 | **0.895** ± 0.001 | 0.889 ± 0.003 ‖ | 0.884 ± 0.001 | **0.890** ± 0.001 | **0.892** ± 0.001 |
| | ece ↓ | **0.013** ± 0.003 | **0.012** ± 0.005 | **0.014** ± 0.003 | **0.010** ± 0.003 ‖ | 0.020 ± 0.001 | **0.016** ± 0.001 | **0.016** ± 0.001 |
| fmnist (lenet) | nll ↓ | 0.232 ± 0.002 | **0.219** ± 0.002 | **0.216** ± 0.002 | 0.226 ± 0.004 ‖ | 0.230 ± 0.005 | 0.224 ± 0.003 | **0.212** ± 0.001 |
| | acc ↑ | 0.919 ± 0.001 | **0.924** ± 0.001 | **0.926** ± 0.002 | 0.920 ± 0.002 ‖ | 0.920 ± 0.001 | 0.920 ± 0.001 | **0.924** ± 0.001 |
| | ece ↓ | **0.017** ± 0.005 | **0.014** ± 0.004 | **0.018** ± 0.002 | **0.013** ± 0.004 ‖ | 0.017 ± 0.002 | 0.015 ± 0.001 | **0.009** ± 0.001 |

the $m$ distributions. Similar to [52], at prediction time, we take $\boldsymbol{\lambda}_k$ to be equal to the means $\boldsymbol{\lambda}_k^{\mathrm{mean}}$ of the distributions $p_t(\boldsymbol{\lambda}_k)$. In Appendix B.3, we provide additional details about $p_t$.

The validation steps (6) and (9) seek to optimize the bounds of the ranges. More specifically, the loss $\ell_{\mathrm{val}}$ favors compact ranges around a good hyperparameter value whereas the entropy term encourages wide ranges, as traded off by $\tau$. We provide an example of the optimization trajectory of $\lambda$ and its range in Figure 2-(right), where $\lambda$ corresponds to the mean of the log-uniform distribution.

# 5 Experiments

Throughout the experiments, we use both metrics that depend on the predictive uncertainty—negative log-likelihood (NLL) and expected calibration error (ECE) [55]—and metrics that do not, e.g., the classification accuracy. The supplementary material also reports Brier score [10] (for which we typically observed a strong correlation with NLL). Moreover, as diversity metric, we take the predictive disagreement of the ensemble members normalized by (1-accuracy), as used in [22]. In the tables, we write the number of ensemble members in brackets "(·)" next to the name of the methods.

## 5.1 Multi-layer perceptron and LeNet on Fashion MNIST & CIFAR-100

To validate our approaches and run numerous ablation studies, we first focus on small-scale models, namely MLP and LeNet [44], over CIFAR-100 [40] and Fashion MNIST [73]. For both models, we add a dropout layer [66] before their last layer. For each pair of dataset/model type, we consider two tuning settings involving the dropout rate and different $L_2$ regularizers defined with varied granularity, e.g., layerwise. Appendix C.1 gives all the details about the training, tuning and dataset definitions.

**Baselines.** We compare our methods (i) `hyper-deep ens`: hyper-deep ensemble of Section 3 and (ii) `hyper-batch ens`: hyper-batch ensemble of Section 4, to (a) `rand search`: the best single model after 50 trials of random search [6], (b) `Bayes opt`: the best single model after 50 trials of Bayesian optimization [63, 27], (c) `deep ens`: deep ensemble [43] using the best hyperparameters found by random search, (d) `batch ens`: batch ensemble [69], (e) `STN`: self-tuning networks [52], and (f) `fixed init hyper ens`: defined in Section 3. The supplementary material details how we tune the hyperparameters specific to `batch ens`, `STN` and `hyper-batch ens` (see Appendix C.2, Appendix C.3 and Appendix C.4 and further ablations about e in Appendix C.5 and $\tau$ in Appendix C.6). Note that `batch ens` needs the tuning of its own hyperparameters and those of the MLP/LeNet models, while `STN` and `hyper-batch ens` automatically tune the latter.

We highlight below the key conclusions from Table 1 with single models and ensemble of sizes 3. The same conclusions can also be drawn for the ensemble of size 5 (see Appendix C.7.1).

Table 2: Performance of ResNet-20 (upper table) and Wide ResNet-28-10 (lower table) models on CIFAR-10/100. We separately compare the *efficient* methods (2 rightmost columns) and we mark in bold the best results (within one standard error). Our two methods hyper-deep/hyper-batch ensembles improve upon deep/batch ensembles.

| (ResNet-20) | | single (1) | deep ens (4) | hyper-deep ens (4) ‖ | batch ens (4) | hyper-batch ens (4) |
|---|---|---|---|---|---|---|
| cifar100 | nll ↓ | 1.178 ± 0.020 | 0.971 ± 0.002 | **0.925** ± 0.002 | 1.235 ± 0.007 | **1.152** ± 0.015 |
| | acc ↑ | 0.682 ± 0.005 | **0.726** ± 0.000 | **0.742** ± 0.001 | 0.697 ± 0.000 | **0.699** ± 0.002 |
| | ece ↓ | 0.064 ± 0.005 | 0.059 ± 0.000 | **0.049** ± 0.001 | 0.119 ± 0.001 | **0.095** ± 0.002 |
| | div ↑ | – | **1.177** ± 0.004 | **1.323** ± 0.001 | **0.154** ± 0.006 | **0.159** ± 0.007 |
| cifar10 | nll ↓ | 0.262 ± 0.006 | **0.193** ± 0.000 | **0.192** ± 0.004 | 0.278 ± 0.004 | **0.235** ± 0.004 |
| | acc ↑ | 0.927 ± 0.001 | 0.937 ± 0.000 | **0.940** ± 0.000 | **0.929** ± 0.000 | **0.929** ± 0.001 |
| | ece ↓ | 0.035 ± 0.001 | **0.010** ± 0.000 | 0.012 ± 0.001 | 0.039 ± 0.001 | **0.017** ± 0.000 |
| | div ↑ | – | 1.393 ± 0.025 | **1.451** ± 0.018 | 0.789 ± 0.010 | **0.821** ± 0.013 |

| (WRN-28-10) | | single (1) | deep ens (4) | hyper-deep ens (4) ‖ | batch ens (4) | hyper-batch ens (4) |
|---|---|---|---|---|---|---|
| cifar100 | nll ↓ | 0.811 ± 0.026 | 0.661 ± 0.001 | **0.652** ± 0.000 | 0.690 ± 0.005 | **0.678** ± 0.005 |
| | acc ↑ | 0.801 ± 0.004 | 0.826 ± 0.001 | **0.828** ± 0.000 | **0.819** ± 0.001 | **0.820** ± 0.000 |
| | ece ↓ | 0.062 ± 0.001 | 0.022 ± 0.000 | **0.019** ± 0.000 | 0.026 ± 0.002 | **0.022** ± 0.001 |
| | div ↑ | – | 0.956 ± 0.009 | **1.086** ± 0.011 | 0.761 ± 0.014 | **0.996** ± 0.015 |
| cifar10 | nll ↓ | 0.152 ± 0.009 | 0.125 ± 0.000 | **0.115** ± 0.001 | 0.136 ± 0.001 | **0.126** ± 0.001 |
| | acc ↑ | 0.961 ± 0.001 | 0.962 ± 0.000 | **0.965** ± 0.001 | **0.963** ± 0.001 | 0.963 ± 0.000 |
| | ece ↓ | 0.023 ± 0.005 | **0.007** ± 0.000 | **0.007** ± 0.000 | 0.017 ± 0.001 | **0.009** ± 0.001 |
| | div ↑ | – | 0.866 ± 0.017 | **1.069** ± 0.025 | 0.444 ± 0.003 | **0.874** ± 0.026 |

**Ensembles benefit from both weight and hyperparameter diversity.** With the pictorial view of Figure 2 in mind, `fixed init hyper ens`, i.e., a "row", tends to outperform `deep ens`, i.e., a "column". Moreover, those two approaches (as well as the other methods of the benchmark) are outperformed by our stratified procedure `hyper-deep ens`, demonstrating the benefit of combining hyperparameter and initialization diversity (see Appendix C.7.2 for the detailed assessment of the statistical significance). In Appendix C.7.3, we study more specifically the diversity and we show that `hyper-deep ens` has indeed more diverse predictions than `deep ens`.

**Efficient ensembles benefit from both weight and hyperparameter diversity.** Among the efficient approaches (the three rightmost columns of Table 1), `hyper-batch ens` performs best. It improves upon both `STN` and `batch ens`, the two methods it builds upon. In line with [52], `STN` typically matches or improves upon `rand search` and `Bayes opt`. As explained in Section 4.1, `hyper-batch ens` has however twice the number of parameters of `batch ens`. In Appendix C.7.4, we thus compare with a "deep ensemble of two batch ensembles" (i.e., resulting in the same number of parameters but twice as many members as for `hyper-batch ens`). In that case, `hyper-batch ens` also either improves upon or matches the performance of the combination of two `batch ens`.

## 5.2 ResNet-20 and Wide ResNet-28-10 on CIFAR-10 & CIFAR-100

We evaluate our approach in a large-scale setting with ResNet-20 [31] and Wide ResNet 28-10 models [74] as they are simple architectures with competitive performance on image classification tasks. We consider six different $L_2$ regularization hyperparameters (one for each block of the ResNet) and a label smoothing hyperparameter. We show results on CIFAR-10, CIFAR-100 and corruptions on CIFAR-10 [33, 64]. Moreover, in Appendix D.3, we provide additional out-of-distribution evaluations along the line of [32]. Further details about the experiment settings can be found in Appendix D.

**CIFAR-10/100.** We compare `hyper-deep ens` with a `single` model (tuned as next explained) and `deep ens` of varying ensemble sizes. Our `hyper-deep ens` is constructed based on 100 trials of random search while `deep ens` and `single` take the best hyperparameter configuration found by the random search procedure. Figure 1 displays the results on CIFAR-100 along with the standard errors and shows that throughout the ensemble sizes, there is a substantial performance improvement of hyper-deep ensembles over deep ensembles. The results for CIFAR-10 are shown in Appendix D where `hyper-deep ens` leads to consistent but smaller improvements, e.g., in terms of NLL. We next fix the ensemble size to four and compare the performance of `hyper-batch ens` with the direct competing method `batch ens`, as well as with `hyper-deep ens`, `deep ens` and `single`.

The results are reported in Table 2. On CIFAR-100, `hyper-batch ens` improves, or matches, `batch ens` across all metrics. For instance, in terms of NLL, it improves upon `batch ens` by about 7% and 2% for ResNet-20 and Wide ResNet 28-10 respectively. Moreover, the members of `hyper-batch ens` make more diverse predictions than those of `batch ens`. On CIFAR-10 `hyper-batch ens` also achieves a consistent improvement, though less pronounced (see Table 2). On the same Wide ResNet 28-10 benchmark, with identical training and evaluation pipelines (see https://github.com/google/uncertainty-baselines), variational inference [70] leads to (NLL, ACC, ECE)=(0.211, 0.947, 0.029) and (NLL, ACC, ECE)=(0.944, 0.778, 0.097) for CIFAR-10 and CIFAR-100 respectively, while Monte Carlo dropout [23] gets (NLL, ACC, ECE)=(0.160, 0.959, 0.024) and (NLL, ACC, ECE)=(0.830, 0.776, 0.050) for CIFAR-10 and CIFAR-100 respectively.

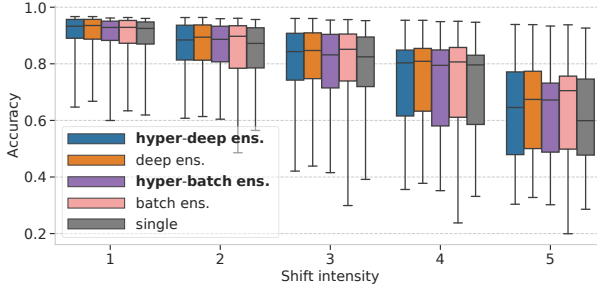

Figure 3: CIFAR-10 corruptions. Each box shows the quartiles summarizing the results across all types of shifts while the error bars give the min/max across different shift types.

We can finally look at how the joint training in `hyper-batch ens` leads to *complementary* ensemble members. For instance, for Wide ResNet 28-10 on CIFAR-100, while the ensemble performance are (NLL, ACC)=(0.678, 0.820) (see Table 2), the individual members obtain substantially poorer performance, as measured by the *average ensemble-member metrics* (NLL, ACC)=(0.904, 0.788).

**Training time and memory cost.** Both in terms of the number of parameters and training time, `hyper-batch ens` is about twice as costly as `batch ens`. For CIFAR-100, `hyper-batch ens` takes 2.16 minutes/epoch and `batch ens` 1.10 minute/epoch. More details are available in Appendix D.6.

**Calibration on out of distribution data.** We measure the calibrated prediction on corrupted datasets, which is a type of out-of-distribution examples. We consider the recently published dataset by [33], which consists of over 30 types of corruptions to the images of CIFAR-10. A similar benchmark can be found in [64]. On Figure 3, we find that all ensembles methods improve upon the single model. The mean accuracies are similar for all ensemble methods, whereas `hyper-batch ens` shows more robustness than `batch ens` as it typically leads to smaller worst values (see bottom whiskers in Figure 3). Plots for calibration error and NLL can be found in Appendix D.5.

# 6 Discussion

We envision several promising directions for future research.

**Towards more compact parametrization.** In this work, we have used the layers from [52] that lead to a 2x increase in memory compared with standard layers. In lieu of (3), *low-rank* parametrizations, e.g., $\mathbf{W} + \sum_{j=1}^{h} e_j(\boldsymbol{\lambda})\mathbf{g}_j \mathbf{h}_j^\top$, would be appealing to reduce the memory footprint of self-tuning networks and hyper-batch ensembles. We formally show in Appendix E that this family of parametrizations is well motivated in the case of shallow models where they enjoy good approximation guarantees.

**Architecture diversity.** Our proposed hyperparameter ensembles provide diversity with respect to hyperparameters related to regularization and optimization. We would like to go further in ensembling very different functions in the search space, such as network width, depth [2], and the choice of residual block. Doing so connects to older work on Bayesian marginalization over structures [37, 1]. More broadly, we can wonder what other *types* of diversity matter to endow deep learning models with better uncertainty estimates?

## Broader Impact

Our work belongs to a broader research effort that tries to quantify the predictive uncertainty for deep neural networks. Those models are known to generalize poorly to small changes to the data while maintaining high confidence in their predictions.

**Who may benefit from this research?** The broader topic of our work is becoming increasingly important in a context where machine learning systems are being deployed in safety-critical fields, e.g., medical diagnosis [54, 49] and self-driving cars [48]. Those examples would benefit from the general technology we contribute to. In those cases, it is essential to be able to reliably trust the uncertainty output by the models before any decision-making process, to possibly escalate uncertain decisions to appropriate human operators.

**Who may be put at disadvantage from this research?** We are not aware of a group of people that may be put at disadvantage as a result of this direct research.

**What are the consequences of failure of the system?** By definition, our research could contribute to aspects of machine-learning systems used in high-risk domains (e.g., we mentioned earlier medical fields and self-driving cars) which involves complex data-driven decision-making processes. Depending on the nature of the application at hand, a failure of the system could lead to extremely negative consequences. A case in point is the recent screening system used by one third of UK government councils to allocate welfare budget. [1]

**Do the task/method leverage biases in the data?** The method we develop in this work is domain-agnostic and does not rely on specific data assumptions. Our method also does not contain components that would prevent its combination with existing fairness or privacy-preserving technologies [4].

## Acknowledgments

We would like to thank Nicolas Le Roux, Alexey Dosovitskiy and Josip Djolonga for insightful discussions at earlier stages of this project. Moreover, we would like to thank Sebastian Nowozin, Klaus-Robert Müller and Balaji Lakshminarayanan for helpful comments on a draft of this paper.

## Footnotes

[1] Link to the corresponding article in The Guardian, October 2019:

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
