[Supplementary Material]

# Supplementary Material:
# Hyperparameter Ensembles for Robustness and Uncertainty Quantification

## A  Further details about fixed init hyper ensembles and hyper-deep ensembles

We recall the procedure from [13] in Algorithm 2. In words, given a pre-defined set of models $\mathcal{M}$ (e.g., the outcome of random search), we greedily grow an ensemble, until some target size $K$ is met, by selecting with replacement the model leading to the best improvement of some score $\mathcal{S}$ such as the validation negative log-likelihood.

The *with-replacement* selection strategy makes it possible to construct ensembles where the contributions of each member is weighted (see Section 2.1 in [13]). To properly account for the fact that there may be multiple times the same model selected, we use ".unique()" in Algorithms 1-2 to correctly count the number of members.

---

**Algorithm 2:** hyper_ens($\mathcal{M}$, $K$) #  Caruana et al. [13]

1  ensemble $\mathcal{E} = \{\ \}$, score $\mathcal{S}(\cdot)$, $\mathcal{S}_{\text{best}} = +\infty$;
2  **while** $|\mathcal{E}.\text{unique}()| \leq K$ **do**
3  $\quad$ $f_{\boldsymbol{\theta}^\star} = \arg\min_{f_{\boldsymbol{\theta}} \in \mathcal{M}} \mathcal{S}(\mathcal{E} \cup \{f_{\boldsymbol{\theta}}\})$;
4  $\quad$ **if** $\mathcal{S}(\mathcal{E} \cup \{f_{\boldsymbol{\theta}^\star}\}) < \mathcal{S}_{\text{best}}$ **then**
5  $\quad\quad$ $\mathcal{E} = \mathcal{E} \cup \{f_{\boldsymbol{\theta}^\star}\}$, $\mathcal{S}_{\text{best}} = \mathcal{S}(\mathcal{E})$;
6  $\quad$ **else**
7  $\quad\quad$ **return** $\mathcal{E}$;
8  $\quad$ **end**
9  **end**
10  **return** $\mathcal{E}$;

---

## B  Further details about hyper-batch ensemble

### B.1  The structure of the convolutional layer

We detail the structure of the (two-dimensional) convolutional layer of hyper-batch ensemble in the case of $K$ ensemble members. Similar to the dense layer presented in Section 4.1, the convolutional layer is obtained by *composing* the layer of batch ensemble [69] and that of self-tuning networks [52].

Let us denote by $\mathbf{K} \in \mathbb{R}^{l \times l \times c_{\text{in}} \times c_{\text{out}}}$ and $\mathbf{b}_k \in \mathbb{R}^{c_{\text{out}}}$ the convolution kernel and the $k$-th member-specific bias term, with $l$ the kernel size, $c_{\text{in}}$ the number of input channels and $c_{\text{out}}$ the number of output channels (also referred to as the number of filters).

For $k \in \{1, \ldots, K\}$, let us consider the following auxiliary vectors $\mathbf{r}_k, \mathbf{u}_k \in \mathbb{R}^{c_{\text{in}}}$ and $\mathbf{s}_k, \mathbf{v}_k \in \mathbb{R}^{c_{\text{out}}}$. For $\boldsymbol{\Delta}$ of the same shape as $\mathbf{K}$ and the embedding $\mathbf{e}(\boldsymbol{\lambda}_k) \in \mathbb{R}^{c_{\text{out}}}$, we have

$$\mathbf{K}_k(\boldsymbol{\lambda}_k) = \mathbf{K} \circ (\mathbf{r}_k \mathbf{s}_k^\top) + [\boldsymbol{\Delta} \circ (\mathbf{u}_k \mathbf{v}_k^\top)] \circ \mathbf{e}(\boldsymbol{\lambda}_k)^\top \tag{10}$$

where the rank-1 factors are understood to be broadcast along the first two dimensions. Similar, for the bias terms, we have

$$\mathbf{b}_k(\boldsymbol{\lambda}_k) = \mathbf{b}_k + \boldsymbol{\delta}_k \circ \mathbf{e}'(\boldsymbol{\lambda}_k) \tag{11}$$

with $\boldsymbol{\delta}_k, \mathbf{e}'(\boldsymbol{\lambda}_k)$ of the same shape as $\mathbf{b}_k$.

Given the form of (10) and (11), we can observe that the conclusions drawn for the dense layer in Section 4.1 also hold for the convolutional layer.

### B.2  Efficient computation of the $L_2$ regularizer

We recall that each ensemble member manipulates its own hyperparameters $\boldsymbol{\lambda}_k \in \mathbb{R}^m$ and, as required by the training procedure in (8), those hyperparameters are sampled as part of the stochastic optimization.

We focus on the example of a given dense layer, with weight matrix $\mathbf{W}_k(\boldsymbol{\lambda}_k)$ and bias term $\mathbf{b}_k(\boldsymbol{\lambda}_k)$, as exposed in Section 4.1.

Let us consider a minibatch of size $b$ for the $K$ ensemble members, i.e., $\{\boldsymbol{\lambda}_{k,i}\}_{k=1}^K$ for $i \in \{1,\dots,b\}$. Moreover, let us introduce the scalar $\nu_{k,i}$ that is equal to the entry in $\boldsymbol{\lambda}_{k,i}$ containing the value of the $L_2$ penalty for the particular dense layer under study.[2]

With that notation, we concentrate on the efficient computation (especially the vectorization with respect to the minibatch dimension) of

$$\frac{1}{bK}\sum_{i=1}^b\sum_{k=1}^K \nu_{k,i}\|\mathbf{W}_k(\boldsymbol{\lambda}_{k,i})\|^2, \tag{12}$$

the case of the bias term following along the same lines. From Section 4.1 we have

$$\mathbf{W}_k(\boldsymbol{\lambda}_{k,i}) = \mathbf{W}\circ(\mathbf{r}_k\mathbf{s}_k^\top) + [\boldsymbol{\Delta}\circ(\mathbf{u}_k\mathbf{v}_k^\top)]\circ \mathbf{e}(\boldsymbol{\lambda}_{k,i})^\top = \mathbf{W}_k + \boldsymbol{\Delta}_k\circ\mathbf{e}_{k,i}^\top$$

which we have simplified by introducing a few additional shorthands. Let us further introduce

$$\langle\nu_k\rangle = \frac{1}{b}\sum_{i=1}^b\nu_{k,i} \text{ and } \langle\nu_k\mathbf{e}_k\rangle = \frac{1}{b}\sum_{i=1}^b\nu_{k,i}\mathbf{e}_{k,i} \text{ and } \langle\nu_k\mathbf{e}_k^2\rangle = \frac{1}{b}\sum_{i=1}^b\nu_{k,i}(\mathbf{e}_{k,i}\circ\mathbf{e}_{k,i}).$$

We then develop $\|\mathbf{W}_k(\boldsymbol{\lambda}_{k,i})\|^2$ into $\|\mathbf{W}_k\|^2 + 2\mathbf{W}_k^\top(\boldsymbol{\Delta}_k\circ\mathbf{e}_{k,i}^\top) + \|\boldsymbol{\Delta}_k\circ\mathbf{e}_{k,i}^\top\|^2$ and plug the decomposition into (12), with $\boldsymbol{\Delta}_k^2 = \boldsymbol{\Delta}_k\circ\boldsymbol{\Delta}_k$, leading to

$$\frac{1}{K}\sum_{k=1}^K\left\{\langle\nu_k\rangle\|\mathbf{W}_k\|^2 + 2\mathbf{W}_k^\top(\boldsymbol{\Delta}_k\circ\langle\nu_k\mathbf{e}_k\rangle^\top) + \sum_{l,l'}(\boldsymbol{\Delta}_k^2)_{l,l'}\langle\nu_k\mathbf{e}_k^2\rangle_{l'}\right\}$$

for which all the remaining operations can be efficiently broadcast.

### B.3 Details about the choice of the distributions $p_t$

We discuss in this section additional details about the choice of the distributions over the hyperparameters $p_t(\boldsymbol{\lambda}_k) = p(\lambda_k|\boldsymbol{\xi}_{k,t})$.

In the experiments of Section 5, we manipulate hyperparameters $\boldsymbol{\lambda}_k$'s that are positive and bounded (e.g., a dropout rate). To simplify the exposition, let us focus momentarily on a single ensemble member ($K = 1$). Let us further consider such a positive, bounded one-dimensional hyperparameter $\lambda \in [a, b]$, with $0 < a < b$, and define $\phi(t) = (b-a)\,\texttt{sigmoid}(t) + a$, with $\phi^{-1}$ its inverse. In that setting, [52] propose to use for $p_t(\lambda) = p(\lambda|\boldsymbol{\xi}_t)$ the following distribution:

$$\lambda|\boldsymbol{\xi}_t \sim \phi\big(\phi^{-1}(\lambda_t) + \varepsilon\big) \text{ with } \varepsilon\sim\mathcal{N}(0,\sigma_t) \text{ and } \boldsymbol{\xi}_t = \{\sigma_t,\lambda_t\}. \tag{13}$$

In preliminary experiments we carried out, we encountered issues with (13), e.g., $\lambda$ consistently pushed to its lower bound $a$ during the optimization.

We have therefore departed from (13) and have focused instead on a simple log-uniform distribution, which is a standard choice for hyperparameter tuning, e.g., [5, 6, 53]. Its probability density function is given by

$$p(\lambda|\boldsymbol{\xi}_t) = 1/(\lambda\log(b/a)) \text{ with } \boldsymbol{\xi}_t = \{a, b\},$$

while its entropy equals $\mathcal{H}[p(\lambda|\boldsymbol{\xi}_t)] = 0.5(\log(a) + \log(b)) + \log(\log(b/a))$. The mean of the distribution is given by $(b-a)/(\log(b) - \log(a))$ and is used to make predictions.

To summarize, and going back to the setting with $K$ ensemble members and $m$-dimensional $\boldsymbol{\lambda}_k$'s, the optimization of $\{\boldsymbol{\xi}_{k,t}\}_{k=1}^K$ in the validation step involves $2mK$ parameters, i.e., the lower/upper bounds for each hyperparameter and for each ensemble member (in practice, $K \approx 5$ and $m \approx 5 - 10$).

## C Further details about the MLP and LeNet experiments

We provide in this section additional material about the experiments based on MLP and LeNet.

## C.1   MLP and LeNet archtectures and experimental settings

The architectures of the models are:

- **MLP**: The multi-layer perceptron is composed of 2 hidden layers with 200 units each. The activation function is ReLU. Moreover a dropout layer is added before the last layer.
- **LeNet** [44]: This convolutional neural network is composed of a first conv2D layer (32 filters) with a max-pooling operation followed by a second conv2D layer (64 filters) with a max-pooling operation and finally followed by two dense layers (512 and number-of-classes units). The activation function is ReLU everywhere. Moreover, we add a dropout layer before the last dense layer.

As briefly discussed in the main paper, in the first tuning setting (i), there are two $L_2$ regularization parameters for those models: one for all the weight matrices and one for all the bias terms of the conv2D/dense layers; in the second tuning setting (ii), the $L_2$ regularization parameters are further split on a per-layer basis (i.e., a total of $3 \times 2 = 6$ and $4 \times 2 = 8$ $L_2$ regularization parameters for MLP and LeNet respectively).

The ranges for the dropout and $L_2$ parameters are $[10^{-3}, 0.9]$ and $[10^{-3}, 10^3]$ across all settings (i)-(ii), models and datasets (CIFAR-100 and Fashion MNIST).

We take the official train/test splits of the two datasets, and we further subdivide (80%/20%) the train split into actual train/validation sets. We use everywhere Adam [39] with learning rate $10^{-4}$, a batchsize of 256 and 200 (resp. 500) training epochs for LeNet (resp. MLP). We tune all methods to minimize the validation NLL. All the experiments are repeated with 3 random seeds.

## C.2   Selection of the hyperparameters of batch ensemble

Following the recommendations from [69], we tuned

- The type of the initialization of the vectors $\mathbf{r}_k$'s and $\mathbf{s}_k$'s (see Section 2.1). We indeed observed that the performance was sensitive to this choice. We selected from the different initialization schemes proposed in [69]
  - Entries distributed according to the Gaussian distribution $\mathcal{N}(\mathbf{1}, 0.5 \times \mathbf{I})$
  - Entries distributed according to the Gaussian distribution $\mathcal{N}(\mathbf{1}, 0.75 \times \mathbf{I})$
  - Random independent signs, with probability of $+1$ equal to $0.5$
  - Random independent signs, with probability of $+1$ equal to $0.75$
- A scale factor $\kappa$ to make it possible to reduce the learning rate applied to the vectors $\mathbf{r}_k$'s and $\mathbf{s}_k$'s. Following [69], we considered the scale factor $\kappa$ in $\{1.0, 0.5\}$.
- Whether to use the Gibbs or ensemble cross-entropy at training time. Early experiments showed that Gibbs cross-entropy was substantially better so that we kept this choice fixed thereafter.
- Whether to regularize the vectors $\mathbf{r}_k$'s and $\mathbf{s}_k$'s. [69] mentioned that the two options perform equally well while we observed in those smaller-scale experiments that batch ensemble could overfit in absence of regularization.

The two batch ensemble-specific hyperparameters above (initialization type and $\kappa$) together with the MLP/LeNet hyperparameters were tuned by 50 trials of random search, separately for each ensemble size (3 and 5) and for each triplet (dataset, model type, tuning setting).

## C.3   Selection of the hyperparameters of self-tuning networks

We re-used as much as possible the hyperparameters and design choices from [52], i.e., 5 warm-up epochs (during which no tuning happens) before starting the alternating scheme (2 training steps followed by 1 tuning step).

For the tuning step, the batch size is taken to be the same as that of the training step (256), while the learning was set to $5 \times 10^{-4}$.

Figure 4: LEFT: Evolution of the validation NLL for different choices of the embedding model $\mathbf{e}(\cdot)$. The validation NLL is averaged over all the datasets (Fashion MNIST/CIFAR 100), model types (MLP/LeNet), tuning settings and random seeds. Zero unit means a linear transformation without hidden layer, while $\{64, 128, 256\}$ units are for a single hidden layer. RIGHT: Evolution of the validation NLL for different values of $\tau$. The validation NLL is averaged over all the datasets (Fashion MNIST/CIFAR 100), model types (MLP/LeNet), tuning settings and random seeds.

We tuned the entropic regularization parameter $\tau \in \{0.01, 0.001, 0.0001\}$, separately for each triplet (dataset, model type, tuning setting), as done for all the methods compared in the benchmark. We observed that $\tau = 0.001$ was often found to be the best option, and it therefore constitutes a good default value, as reported in [52].

As studied in Appendix C.5, we fix the embedding model $\mathbf{e}(\cdot)$ to be an MLP with one hidden layer of 64 units and a tanh activation.

## C.4 Selection of the hyperparameters of hyper-batch ensemble

We followed the very same protocol as that used for the standard self-tuning network (as described in Appendix C.3).

By construction, we also inherit from the batch ensemble-specific hyperparameters (see Appendix C.2). To keep the protocol simple, we only tune the most important hyperparameter, namely the type of the initialization of the rank-1 terms (while the scale factor $\kappa$ to discount the learning rate was not considered). As for any other methods in the benchmark, $\tau$ and the initialization type were tuned separately for each triplet (dataset, model type, tuning setting).

For good default choices, we recommend to take $\tau = 0.001$ and use an initialization scheme with random independent signs (with the probability of $+1$ equal to $0.75$).

## C.5 Choice of the embedding $\mathbf{e}(\cdot)$

We study the impact of the choice of the model that defines the embedding $\mathbf{e}(\cdot)$.

In [52], $\mathbf{e}(\cdot)$ is taken to be a simple linear transformation. In a slightly different context, the authors of [16] consider MLPs with one hidden layer of 128 or 256 units, depending on their applications.

In the light of those previous choices, we compare the performance of different architectures of $\mathbf{e}(\cdot)$, namely linear (i.e., 0 units) and one hidden layer of 64, 128, and 256 units. The results are summarized in Figure 4-(left), for different ensemble sizes (one corresponding to the standard self-tuning networks [52]). We computed the validation NLL averaged over all the datasets (Fashion MNIST/CIFAR 100), model types (MLP/LeNet), tuning settings and random seeds.

Based on Figure 4-(left), we select for $\mathbf{e}(\cdot)$ an MLP with a single hidden layer of 64 units and a tanh activation function.

## C.6 Sensitivity analysis with respect to the entropy regularization parameter $\tau$

We study the impact of the choice of the entropy regularization parameter $\tau$ in (9). We report in Figure 4-(right) how the validation negative log-likelihood—aggregated over all the datasets

Table 3: Comparison over CIFAR 100 and Fashion MNIST with MLP and LeNet architectures. The table reports means ± standard errors (over the 3 random seeds and pooled over the 2 tuning settings), for ensemble approaches with 3 and 5 members (the *efficient* approaches are compared separately in Table 4). "fixed init ens" is a shorthand for `fixed init hyper ens`, i.e., a "row" in Figure 2-(left). Our method hyper-deep ensemble improves upon deep ensemble (in Appendix C.7.2, we assess the statistical significance of those improvements with a Wilcoxon signed-rank test, paired along settings, datasets and model types).

| | | fixed init ens (3) | fixed init ens (5) | hyper-deep ens (3) | hyper-deep ens (5) | deep ens (3) | deep ens (5) |
|---|---|---|---|---|---|---|---|
| cifar100 (mlp) | nll ↓ | **2.943** ± 0.010 | **2.920** ± 0.007 | 2.953 ± 0.058 | **2.919** ± 0.041 | **2.969** ± 0.057 | **2.946** ± 0.041 |
| | acc ↑ | 0.287 ± 0.003 | **0.292** ± 0.002 | 0.291 ± 0.004 | **0.296** ± 0.003 | 0.289 ± 0.003 | **0.292** ± 0.004 |
| | brier ↓ | -0.161 ± 0.002 | -0.165 ± 0.001 | -0.164 ± 0.003 | **-0.169** ± 0.002 | -0.160 ± 0.004 | -0.163 ± 0.003 |
| | ece ↓ | **0.029** ± 0.007 | **0.025** ± 0.006 | 0.022 ± 0.007 | 0.023 ± 0.005 | 0.038 ± 0.014 | 0.035 ± 0.007 |
| cifar100 (lenet) | nll ↓ | **2.259** ± 0.067 | **2.248** ± 0.069 | 2.211 ± 0.066 | **2.136** ± 0.057 | **2.334** ± 0.141 | **2.298** ± 0.146 |
| | acc ↑ | 0.439 ± 0.008 | 0.445 ± 0.010 | 0.452 ± 0.007 | **0.466** ± 0.006 | 0.421 ± 0.026 | 0.428 ± 0.027 |
| | brier ↓ | -0.301 ± 0.010 | **-0.305** ± 0.012 | -0.315 ± 0.010 | **-0.330** ± 0.008 | -0.282 ± 0.030 | -0.288 ± 0.031 |
| | ece ↓ | **0.049** ± 0.023 | **0.045** ± 0.021 | 0.039 ± 0.013 | **0.034** ± 0.008 | 0.050 ± 0.015 | 0.045 ± 0.022 |
| fmnist (mlp) | nll ↓ | 0.312 ± 0.003 | **0.305** ± 0.003 | 0.310 ± 0.001 | **0.305** ± 0.001 | 0.319 ± 0.005 | 0.318 ± 0.006 |
| | acc ↑ | 0.893 ± 0.001 | **0.897** ± 0.000 | 0.895 ± 0.001 | **0.897** ± 0.000 | 0.889 ± 0.003 | 0.889 ± 0.003 |
| | brier ↓ | -0.843 ± 0.001 | **-0.848** ± 0.001 | -0.845 ± 0.001 | **-0.848** ± 0.001 | -0.839 ± 0.003 | -0.840 ± 0.003 |
| | ece ↓ | **0.012** ± 0.005 | **0.014** ± 0.002 | 0.014 ± 0.003 | 0.017 ± 0.001 | 0.010 ± 0.003 | 0.009 ± 0.003 |
| fmnist (lenet) | nll ↓ | 0.219 ± 0.002 | 0.215 ± 0.002 | 0.216 ± 0.002 | **0.210** ± 0.002 | 0.226 ± 0.004 | 0.222 ± 0.005 |
| | acc ↑ | 0.924 ± 0.001 | **0.926** ± 0.001 | **0.926** ± 0.002 | **0.928** ± 0.001 | 0.920 ± 0.002 | 0.921 ± 0.005 |
| | brier ↓ | -0.889 ± 0.001 | **-0.891** ± 0.001 | **-0.890** ± 0.002 | **-0.893** ± 0.001 | -0.883 ± 0.003 | -0.884 ± 0.003 |
| | ece ↓ | **0.014** ± 0.004 | **0.015** ± 0.002 | 0.018 ± 0.002 | **0.014** ± 0.003 | **0.013** ± 0.004 | **0.011** ± 0.003 |

(Fashion MNIST/CIFAR 100), model types (MLP/LeNet), tuning settings and random seeds—varies with $\tau \in \{0.01, 0.001, 0.0001\}$.

As discussed in Appendix C.3 and in Appendix C.2, a good default value, as already reported in [52] is $\tau = 0.001$.

## C.7 Complementary results

### C.7.1 Results for ensembles of size 3 and 5

In Table 3 and Table 4 (the latter table contains the efficient ensemble methods), we complete Table 9 with the addition of the results for the ensembles of size 5. To ease the comparison across different ensemble sizes, we incorporate as well the results for the size 3.

The conclusions highlighted in the main paper also hold for the larger ensembles of size 5. In Table 4, we can observe that `hyper-batch ens` with 5 members does not consistently improve upon its counterpart with 3 members. This trend is corrected if more training epochs are considered (see in Table 7 the effect of twice as many training epochs).

### C.7.2 Assessment of the statistical significance of the results

To assess the statistical significance of the improvements displayed in Table 1, Table 3 and Table 4, we run the Wilcoxon signed-rank test, paired along settings, datasets and model types. We report the results in Table 5. The pairing of the tests is especially important for the comparisons between `deep ens`, `fixed init hyper ens` and `hyper-deep ens` since their respective performances are heavily conditioned on the initial random searches they build upon.

First, we can see that `hyper-deep ens` significantly improves upon both `deep ens` and `fixed init hyper ens` (with larger p-values in the latter case, though). Second, while `hyper-batch ens` significantly improves upon STN, `hyper-batch ens` can only be shown to be better than `batch ens` in terms of likelihood (with a 5% significance level). Overall, we also observe that we do not have significant improvements with respect to ECE which is known to be more noisy [57].

Table 4: Comparison of the *efficient* ensemble methods over CIFAR 100 and Fashion MNIST with MLP and LeNet architectures. The table reports means ± standard errors (over the 3 random seeds and pooled over the 2 tuning settings), for ensemble approaches with 3 and 5 members. Our method hyper-batch ensemble improves upon batch ensemble (in Appendix C.7.2, we assess the statistical significance of those improvements with a Wilcoxon signed-rank test, paired along settings, datasets and model types).

| | | hyper-batch ens (3) | hyper-batch ens (5) | batch ens (3) | batch ens (5) |
|---|---|---|---|---|---|
| cifar100 (mlp) | nll ↓ | **2.979** ± 0.004 | **2.983** ± 0.001 | 3.015 ± 0.003 | 3.056 ± 0.004 |
| | acc ↑ | **0.281** ± 0.002 | **0.282** ± 0.001 | 0.275 ± 0.001 | 0.265 ± 0.001 |
| | brier ↓ | **-0.157** ± 0.000 | **-0.157** ± 0.000 | -0.153 ± 0.001 | -0.141 ± 0.000 |
| | ece ↓ | 0.030 ± 0.002 | 0.034 ± 0.001 | **0.022** ± 0.002 | 0.033 ± 0.002 |
| cifar100 (lenet) | nll ↓ | **2.283** ± 0.016 | **2.297** ± 0.009 | **2.350** ± 0.024 | **2.239** ± 0.027 |
| | acc ↑ | 0.428 ± 0.003 | 0.425 ± 0.002 | **0.438** ± 0.003 | **0.437** ± 0.006 |
| | brier ↓ | **-0.288** ± 0.003 | -0.282 ± 0.002 | **-0.295** ± 0.003 | **-0.296** ± 0.008 |
| | ece ↓ | **0.058** ± 0.004 | 0.069 ± 0.006 | **0.058** ± 0.015 | **0.038** ± 0.018 |
| fmnist (mlp) | nll ↓ | 0.308 ± 0.002 | **0.304** ± 0.001 | 0.351 ± 0.004 | 0.320 ± 0.002 |
| | acc ↑ | **0.892** ± 0.001 | **0.892** ± 0.001 | 0.884 ± 0.001 | **0.892** ± 0.001 |
| | brier ↓ | **-0.844** ± 0.001 | **-0.845** ± 0.001 | -0.830 ± 0.001 | **-0.844** ± 0.001 |
| | ece ↓ | 0.016 ± 0.001 | **0.013** ± 0.001 | 0.020 ± 0.001 | 0.024 ± 0.001 |
| fmnist (lenet) | nll ↓ | **0.212** ± 0.001 | **0.209** ± 0.002 | 0.230 ± 0.005 | 0.221 ± 0.002 |
| | acc ↑ | **0.924** ± 0.001 | **0.925** ± 0.001 | 0.920 ± 0.001 | 0.922 ± 0.001 |
| | brier ↓ | **-0.889** ± 0.001 | **-0.891** ± 0.001 | -0.883 ± 0.001 | -0.886 ± 0.001 |
| | ece ↓ | **0.009** ± 0.001 | **0.008** ± 0.001 | 0.017 ± 0.002 | 0.015 ± 0.001 |

Table 5: Results of the one-sided, Wilcoxon signed-rank test, paired along settings, datasets and model types. We report the p-values corresponding to the hypothesis that our method (in blue) has worse value than the corresponding competing methods.

| | ens size | p-value (nll) | p-value (acc) | p-value (ece) | ens size | p-value (nll) | p-value (acc) | p-value (ece) |
|---|---|---|---|---|---|---|---|---|
| deep ens ↔ hyper-deep ens | 3 | $1.1 \times 10^{-5}$ | $2.1 \times 10^{-5}$ | 0.25 | 5 | $9.1 \times 10^{-6}$ | $1.9 \times 10^{-5}$ | 0.33 |
| fixed init hyper ens ↔ hyper-deep ens | 3 | 0.0725 | 0.0017 | 0.43 | 5 | 0.0088 | 0.0018 | 0.44 |
| batch ens ↔ hyper-batch ens | 3 | $6.4 \times 10^{-5}$ | 0.13 | 0.31 | 5 | 0.038 | 0.22 | 0.39 |
| STN ↔ hyper-batch ens | 3 | $9.1 \times 10^{-6}$ | $2.6 \times 10^{-5}$ | 0.23 | 5 | $4.5 \times 10^{-5}$ | $1.3 \times 10^{-5}$ | 0.33 |

Table 6: Normalized predictive disagreement from [22] compared over CIFAR 100 and Fashion MNIST with MLP and LeNet architectures. Higher values mean more diversity in the ensemble predictions. The table reports means $\pm$ standard errors (over the 3 random seeds and pooled over the 2 tuning settings), for ensemble approaches with 3 and 5 members.

| | deep ens (3) | deep ens (5) | hyper-deep ens (3) | hyper-deep ens (5) | batch ens (3) | batch ens (5) | hyper-batch ens (3) | hyper-batch ens (5) |
|---|---|---|---|---|---|---|---|---|
| cifar100 (mlp) | $0.570 \pm 0.099$ | $0.573 \pm 0.103$ | $\mathbf{0.707} \pm 0.072$ | $\mathbf{0.732} \pm 0.055$ | $0.700 \pm 0.003$ | $0.453 \pm 0.010$ | $0.765 \pm 0.004$ | $\mathbf{0.841} \pm 0.004$ |
| cifar100 (lenet) | $0.688 \pm 0.107$ | $0.695 \pm 0.114$ | $\mathbf{0.896} \pm 0.045$ | $\mathbf{0.896} \pm 0.038$ | $\mathbf{0.692} \pm 0.028$ | $0.583 \pm 0.034$ | $\mathbf{0.716} \pm 0.005$ | $0.479 \pm 0.011$ |
| fmnist (mlp) | $0.461 \pm 0.063$ | $0.457 \pm 0.040$ | $0.588 \pm 0.046$ | $\mathbf{0.702} \pm 0.057$ | $0.490 \pm 0.014$ | $\mathbf{0.716} \pm 0.003$ | $0.509 \pm 0.009$ | $0.573 \pm 0.008$ |
| fmnist (lenet) | $0.475 \pm 0.057$ | $0.479 \pm 0.060$ | $\mathbf{0.594} \pm 0.041$ | $\mathbf{0.656} \pm 0.043$ | $0.481 \pm 0.047$ | $\mathbf{0.647} \pm 0.015$ | $0.446 \pm 0.015$ | $0.487 \pm 0.008$ |

### C.7.3 Diversity analysis

In this section, we study the diversity of the predictions made by the ensemble approaches from the experiments of Section 5.1.

To this end, we use the predictive disagreement metric from [22]. This metric is based on the average of the pairwise comparisons of the predictions across the ensemble members. For a given pair of members, it is zero when they are making identical predictions, and one when all their predictions differ. We also normalize the diversity metric by the error rate (i.e., one minus the accuracy) to avoid the case where random predictions provide the best diversity.

For ensemble sizes 3 and 5, we compare in Table 6 the approaches hyper-deep ensemble, deep ensemble, hyper-batch ensemble and batch ensemble with respect to this metric. We can draw the following conclusions:

- **hyper-deep ensemble vs. deep ensemble:** Compared to deep ensemble, we can observe that hyper-deep ensemble leads to significantly more diverse predictions, across all combination of (dataset, model type) and ensemble sizes. Moreover, we can also see that the diversity only slightly increases for deep ensemble going from 3 to 5 members, while it increases more markedly for hyper-deep ensemble. We hypothesise this is due to the more diverse set of models (with varied initialization and hyperparameters) that hyper-deep ensemble can tap into.

- **hyper-batch ensemble vs. batch ensemble:** The first observation is that in this setting (the observation turns out to be different in the case of the Wide Resnet 28-10 experiments), batch ensemble leads to the largest diversity in predictions compared to all the other methods. Although lower compared with batch ensemble, the diversity of hyper-batch ensemble is typically higher than, or competitive with the diversity of deep ensembles.

Table 7: Comparison of batch hyperparameter ensemble and batch ensemble over CIFAR 100 and Fashion MNIST with MLP and LeNet models, while accounting for the number of parameters. The table reports means ± standard errors (over the 3 random seeds and pooled over the 2 tuning settings), for ensemble approaches with 3 and 5 members. "2x-" indicates the method benefited from twice as many training epochs. The two rightmost columns correspond to the combination of two batch ensemble models with 3 and 5 members, resulting in 6 and 10 members.

| | | hyper-batch ens (3) | hyper-batch ens (5) | 2x-hyper-batch ens (3) | 2x-hyper-batch ens (5) | batch ens (3×2) | batch ens (5×2) |
|---|---|---|---|---|---|---|---|
| cifar100 (mlp) | nll ↓ | 2.979 ± 0.004 | 2.983 ± 0.001 | 2.974 ± 0.006 | **2.950** ± 0.003 | 2.980 ± 0.002 | 3.031 ± 0.002 |
| | acc ↑ | **0.281** ± 0.002 | **0.282** ± 0.001 | 0.277 ± 0.003 | **0.284** ± 0.002 | **0.282** ± 0.001 | 0.268 ± 0.001 |
| | brier ↓ | -0.157 ± 0.000 | -0.157 ± 0.000 | -0.153 ± 0.001 | **-0.159** ± 0.001 | -0.157 ± 0.000 | -0.144 ± 0.000 |
| | ece ↓ | **0.030** ± 0.002 | **0.034** ± 0.001 | 0.033 ± 0.004 | **0.034** ± 0.005 | 0.032 ± 0.001 | 0.040 ± 0.002 |
| cifar100 (lenet) | nll ↓ | 2.283 ± 0.016 | 2.297 ± 0.009 | 2.255 ± 0.014 | 2.269 ± 0.006 | 2.188 ± 0.008 | **2.163** ± 0.012 |
| | acc ↑ | 0.428 ± 0.003 | 0.425 ± 0.002 | 0.430 ± 0.003 | 0.428 ± 0.001 | **0.460** ± 0.002 | 0.451 ± 0.003 |
| | brier ↓ | -0.288 ± 0.003 | -0.282 ± 0.002 | -0.295 ± 0.002 | -0.291 ± 0.001 | **-0.321** ± 0.001 | -0.309 ± 0.004 |
| | ece ↓ | 0.058 ± 0.004 | 0.069 ± 0.006 | 0.028 ± 0.001 | 0.036 ± 0.006 | **0.017** ± 0.004 | 0.060 ± 0.009 |
| fmnist (mlp) | nll ↓ | 0.308 ± 0.002 | 0.304 ± 0.001 | 0.307 ± 0.001 | **0.303** ± 0.001 | 0.333 ± 0.003 | 0.308 ± 0.001 |
| | acc ↑ | 0.892 ± 0.001 | 0.892 ± 0.001 | **0.893** ± 0.001 | **0.894** ± 0.001 | 0.887 ± 0.001 | **0.894** ± 0.001 |
| | brier ↓ | -0.844 ± 0.001 | **-0.845** ± 0.001 | **-0.845** ± 0.001 | **-0.847** ± 0.001 | -0.836 ± 0.001 | **-0.847** ± 0.000 |
| | ece ↓ | 0.016 ± 0.001 | **0.013** ± 0.001 | 0.015 ± 0.001 | **0.013** ± 0.001 | 0.016 ± 0.001 | 0.020 ± 0.001 |
| fmnist (lenet) | nll ↓ | 0.212 ± 0.001 | **0.209** ± 0.002 | **0.211** ± 0.002 | **0.209** ± 0.001 | 0.220 ± 0.001 | 0.213 ± 0.001 |
| | acc ↑ | **0.924** ± 0.001 | **0.925** ± 0.001 | **0.925** ± 0.001 | **0.925** ± 0.000 | 0.922 ± 0.000 | **0.923** ± 0.001 |
| | brier ↓ | **-0.889** ± 0.001 | **-0.891** ± 0.001 | **-0.890** ± 0.001 | **-0.891** ± 0.001 | -0.887 ± 0.000 | -0.889 ± 0.001 |
| | ece ↓ | **0.009** ± 0.001 | **0.008** ± 0.001 | 0.013 ± 0.001 | 0.012 ± 0.001 | 0.013 ± 0.001 | 0.011 ± 0.001 |

### C.7.4  Further comparison between batch ensemble and hyper-batch ensemble

As described in Section 4.1, the structure of the layers of `hyper-batch ens` leads to a 2x increase in memory compared with standard `batch ens`.

In an attempt to fairly account for this difference in memory footprints, we combine two batch ensemble models trained separately and whose total memory footprint amounts to that of `hyper-batch ens`. This procedure leads to ensembles with 6 and 10 members to compare to `hyper-batch ens` instantiated with 3 and 5 members respectively. To also normalize the training budget, `hyper-batch ens` is given twice as many training epochs as each of the `batch ens` models.

Table 7 presents the results of that comparison. In an nutshell, `hyper-batch ens` either continues to improve upon, or remain competitive with, `batch ens`, while still having the advantage of automatically tuning the hyperparameters of the underlying model (MLP or LeNet).

### C.7.5  Ablation study about hyper-deep ensemble

In this section, we conduct two ablation studies about hyper-deep ensemble to better understand its components. We first focus on the effect of using the greedy algorithm of [13] compared with the top-$K$ procedure used in [60]. Second, we relate Algorithm 1 to the NES-RS procedure concurrently proposed by [75].

**Greedy [13] versus top-$K$ selection?**  Starting from the set of models generated by random search (according to the setting of Section 5.1), we apply both the greedy and top-$K$ selection strategies, as previously used in [60], to form ensembles of size 5. We report the results of the evaluations of those strategies in Figure 5.

We can observe that the greedy procedure outperforms the top-$K$ procedure. While the former has an objective aware of the *ensemble performance*, the latter selects the models based only on their individual performance.

**More models from random search versus fewer models with stratification?**  We still focus on the setting of Section 5.1, with ensembles of size 3 and 5. We study the value of the stratification step in Algorithm 1. To this end, we consider the following comparison that accounts for the total number of trained models:

**(A)** `hyper ens (70)`: Random search with 70 models followed by the greedy procedure of [13]. Note that there is no stratification step in this variant. The resulting method falls back to NES-RS from [75] where the architecture is kept fixed while hyperparameters are varied.

Figure 5: Test accuracy evaluated over CIFAR 100 (LEFT) and Fashion MNIST (RIGHT) for both MLP and LeNet models, when using the greedy and top-$K$ selection strategies to construct ensembles with 5 members. The accuracy is averaged over tuning settings and random seeds.

Table 8: Study of the impact of the stratification when accounting for the total number of models to train. `hyper-deep ens` uses stratification while `hyper ens (70)` does not. The comparison is over CIFAR 100 and Fashion MNIST with MLP and LeNet models. The table reports means $\pm$ standard errors (over the 3 random seeds and pooled over the 2 tuning settings).

|  | ens size | cifar100 (lenet) | cifar100 (mlp) | fmnist (lenet) | fmnist (mlp) |
|---|---|---|---|---|---|
| hyper ens (70) | 3 | ce: $2.214 \pm 0.054$ acc: $0.451 \pm 0.006$ ece: $0.039 \pm 0.009$ | ce: $2.957 \pm 0.047$ acc: $0.291 \pm 0.002$ ece: $0.033 \pm 0.008$ | ce: $0.216 \pm 0.003$ acc: $0.926 \pm 0.001$ ece: $0.016 \pm 0.003$ | ce: $0.310 \pm 0.003$ acc: $0.894 \pm 0.001$ ece: $0.015 \pm 0.002$ |
| hyper-deep ens | 3 | ce: $2.211 \pm 0.066$ acc: $0.452 \pm 0.007$ ece: $0.039 \pm 0.013$ | ce: $2.953 \pm 0.058$ acc: $0.291 \pm 0.004$ ece: $0.022 \pm 0.007$ | ce: $0.216 \pm 0.002$ acc: $0.926 \pm 0.002$ ece: $0.018 \pm 0.002$ | ce: $0.310 \pm 0.001$ acc: $0.895 \pm 0.001$ ece: $0.014 \pm 0.003$ |
|  | ens size | cifar100 (lenet) | cifar100 (mlp) | fmnist (lenet) | fmnist (mlp) |
| hyper ens (70) | 5 | ce: $2.182 \pm 0.053$ acc: $0.459 \pm 0.005$ ece: $0.033 \pm 0.005$ | ce: $2.924 \pm 0.035$ acc: $0.297 \pm 0.002$ ece: $0.024 \pm 0.004$ | ce: $0.210 \pm 0.001$ acc: $0.928 \pm 0.001$ ece: $0.014 \pm 0.002$ | ce: $0.305 \pm 0.002$ acc: $0.897 \pm 0.001$ ece: $0.018 \pm 0.004$ |
| hyper-deep ens | 5 | ce: $2.136 \pm 0.057$ acc: $0.466 \pm 0.006$ ece: $0.034 \pm 0.008$ | ce: $2.919 \pm 0.041$ acc: $0.296 \pm 0.003$ ece: $0.023 \pm 0.005$ | ce: $0.210 \pm 0.002$ acc: $0.928 \pm 0.001$ ece: $0.014 \pm 0.003$ | ce: $0.305 \pm 0.001$ acc: $0.897 \pm 0.000$ ece: $0.017 \pm 0.001$ |

(B) `hyper-deep ens`: The procedure described in Algorithm 1 that uses stratification and starts from 50 models obtained by random search (as used in the experiments of Section 5.1). Note that even though we need to stratify 5 models with 5 seeds, i.e., $5^2$=25 models, we can reuse 5 models from the initial random search so that the total budget is 50+20=70 models to train (plus the cost of the calls to the greedy algorithm which is assumed negligible). The two approaches (A)-(B) therefore involve the same number of models to train.

The results of the comparison are reported in Table 8. While `hyper-deep ens` works slightly better, the differences with `hyper ens (70)` are not substantial. In the setting of Section 5.1, it thus appears that, provided that the initial random search produces enough models, the stratification step may be bypassed. In practice, this scheme, without stratification, can also be more convenient to implement.

### C.7.6 Addendum to the results of Table 1

In Table 9, we complete the results of Table 1 with the addition of the Brier scores. Moreover, we provide the details of the performance of `rand search` and `Bayes opt` since only their aggregated best results were reported in Table 1.

Table 9: Comparison over CIFAR 100 and Fashion MNIST with MLP and LeNet architectures. The table reports means ± standard errors (over the 3 random seeds and pooled over the 2 tuning settings). "fixed init ens" is a shorthand for `fixed init hyper ens`, i.e., a "row" in Figure 2-(left). We separately compare the *efficient* methods (3 rightmost columns) and we mark in bold the best results (within one standard error). Our two methods hyper-deep/hyper-batch ensembles improve upon deep/batch ensembles respectively (in Appendix C.7.2, we assess the statistical significance of those improvements with a Wilcoxon signed-rank test, paired by settings, datasets and model types).

| | | rand search (1) | Bayes opt (1) | fixed init ens (3) | hyper-deep ens (3) | deep ens (3) | batch ens (3) | STN (1) | hyper-batch ens (3) |
|---|---|---|---|---|---|---|---|---|---|
| cifar100 (mlp) | nll ↓ | **3.082** ± 0.127 | 2.977 ± 0.010 | **2.943** ± 0.010 | **2.953** ± 0.058 | **2.969** ± 0.057 | 3.015 ± 0.003 | 3.029 ± 0.006 | **2.979** ± 0.004 |
| | acc ↑ | 0.272 ± 0.003 | 0.277 ± 0.002 | **0.287** ± 0.003 | **0.291** ± 0.004 | **0.289** ± 0.003 | 0.275 ± 0.001 | 0.268 ± 0.002 | **0.281** ± 0.002 |
| | brier ↓ | -0.142 ± 0.016 | -0.152 ± 0.003 | **-0.161** ± 0.002 | **-0.164** ± 0.003 | **-0.160** ± 0.004 | -0.153 ± 0.001 | -0.145 ± 0.001 | **-0.157** ± 0.000 |
| | ece ↓ | **0.048** ± 0.037 | **0.034** ± 0.008 | **0.029** ± 0.007 | **0.022** ± 0.007 | **0.038** ± 0.014 | **0.022** ± 0.002 | 0.033 ± 0.004 | 0.030 ± 0.002 |
| cifar100 (lenet) | nll ↓ | 2.523 ± 0.140 | **2.399** ± 0.204 | **2.259** ± 0.067 | **2.211** ± 0.066 | **2.334** ± 0.141 | 2.350 ± 0.024 | 2.329 ± 0.017 | **2.283** ± 0.016 |
| | acc ↑ | 0.395 ± 0.026 | 0.420 ± 0.011 | **0.439** ± 0.008 | **0.452** ± 0.007 | **0.421** ± 0.026 | **0.438** ± 0.003 | 0.415 ± 0.003 | 0.428 ± 0.003 |
| | brier ↓ | -0.249 ± 0.028 | -0.270 ± 0.029 | **-0.301** ± 0.010 | **-0.315** ± 0.010 | **-0.282** ± 0.030 | **-0.295** ± 0.003 | -0.280 ± 0.002 | -0.288 ± 0.003 |
| | ece ↓ | **0.064** ± 0.036 | **0.071** ± 0.054 | **0.049** ± 0.023 | **0.039** ± 0.013 | **0.050** ± 0.015 | 0.058 ± 0.015 | **0.024** ± 0.007 | 0.058 ± 0.004 |
| fmnist (mlp) | nll ↓ | 0.327 ± 0.005 | 0.323 ± 0.003 | **0.312** ± 0.003 | **0.310** ± 0.001 | 0.319 ± 0.005 | 0.351 ± 0.004 | 0.316 ± 0.003 | **0.308** ± 0.002 |
| | acc ↑ | 0.888 ± 0.002 | 0.889 ± 0.002 | **0.893** ± 0.001 | **0.895** ± 0.001 | 0.889 ± 0.003 | 0.884 ± 0.001 | **0.890** ± 0.001 | **0.892** ± 0.001 |
| | brier ↓ | -0.836 ± 0.003 | -0.838 ± 0.002 | **-0.843** ± 0.001 | **-0.845** ± 0.001 | -0.839 ± 0.003 | -0.830 ± 0.001 | -0.840 ± 0.002 | **-0.844** ± 0.001 |
| | ece ↓ | **0.013** ± 0.003 | 0.022 ± 0.004 | **0.012** ± 0.005 | **0.014** ± 0.003 | **0.010** ± 0.003 | 0.020 ± 0.001 | **0.016** ± 0.001 | **0.016** ± 0.001 |
| fmnist (lenet) | nll ↓ | 0.232 ± 0.002 | 0.237 ± 0.002 | **0.219** ± 0.002 | **0.216** ± 0.002 | 0.226 ± 0.004 | 0.230 ± 0.005 | 0.224 ± 0.003 | **0.212** ± 0.001 |
| | acc ↑ | 0.919 ± 0.001 | 0.918 ± 0.002 | **0.924** ± 0.001 | **0.926** ± 0.002 | 0.920 ± 0.002 | 0.920 ± 0.001 | 0.920 ± 0.001 | **0.924** ± 0.001 |
| | brier ↓ | -0.881 ± 0.001 | -0.879 ± 0.002 | -0.889 ± 0.001 | -0.890 ± 0.002 | -0.883 ± 0.003 | -0.883 ± 0.001 | -0.884 ± 0.001 | **-0.889** ± 0.001 |
| | ece ↓ | **0.019** ± 0.004 | **0.017** ± 0.005 | **0.014** ± 0.004 | **0.018** ± 0.002 | **0.013** ± 0.004 | 0.017 ± 0.002 | 0.015 ± 0.001 | **0.009** ± 0.001 |

# D Further details about the ResNet experiments

## D.1 Details about the optimization methods

We first explain the setting we used for training the ResNet 20 and Wide ResNet 28-10 architectures in Section 5 and conclude with the results of an empirical study over different algorithmic choices.

**Training and model definition.** In the following we present the details for our training procedures. A similar training setup to ours for `batch ens` based on a Wide ResNet architecture can be found in the `uncertainty-baselines` repository[3].

For all methods (`hyper-batch ens`, `batch ens`, `hyper-deep ens` and `deep ens`), we optimize the model parameters using stochastic gradient descent (SGD) with Nesterov momentum of $0.9$. For the ResNet 20 model we decay the learning rate by a factor of $0.1$ after the epochs $\{80, 180, 200\}$ and for the Wide ResNet 28-10 model by a factor of $0.2$ after the epochs $\{100, 200, 225\}$. For tuning the hyperparameters in `hyper-batch ens`, we use Adam [39] with a fixed learning rate. For `hyper-batch ens`, we use 95% of the data for training and the remaining 5% for optimizing the hyperparameters $\boldsymbol{\lambda}$ in the tuning step. For the other methods we use the full training set.

For the efficient ensemble methods (`hyper-batch ens` and `batch ens`), we initialize the rank-1 factors, i.e., $\mathbf{r}_k\mathbf{s}_k^\top$ and $\mathbf{u}_k\mathbf{v}_k^\top$ in (7), with entries independently sampled according to $\mathcal{N}(1, 0.5)$ for ResNet 20 and sampled according to $\mathcal{N}(1, 1)$ for Wide ResNet 28-10.

We make two minor adjustments of our model to adapt to the specific structure of the highly overparametrized ResNet models. First, we find that *coupling* the rank-1 factors corresponding to the hyperparamters to the rank-1 factors of weights is beneficial, i.e. we set $\mathbf{u}_k := \mathbf{r}_k$ and $\mathbf{v}_k := \mathbf{s}_k$. This slightly decreases the flexibility of `hyper-batch ens` and makes it more robust against overfitting.

Second, we exclude the rank-1 factors from being regularized. In the original paper introducing `batch ens` [69], the authors mention that both options were found to work equally well and they finally choose *not to regularize* the rank-1 factors (to save extra computation). In our setting, we observe that this choice is important and regularizing the rank-1 factors leads to worse performance (a detailed analysis is given in Appendix D.2). Hence, we do *not* include the rank-1 factors in the regularization.

Table 10: Wide ResNet 28-10. Ablation for including the rank-1 factors of the efficient ensemble methods into the regularization. We run a grid search over all optimization parameters outlined in Appendix D.1 and report the mean performance on CIFAR-100 along with the standard error as well as the best performance attained by all configurations considered. Regularizing the factors substantially decreases the performance of both methods. The results for the unregularized version can be found in the main text, in Table 2.

|  | Mean acc. | Max. acc. | Mean NLL | Min. NLL |
|---|---|---|---|---|
| `hyper-batch ens` | $0.797 \pm 0.004$ | $0.802$ | $0.783 \pm 0.023$ | $0.750$ |
| `batch ens` | $0.797 \pm 0.004$ | $0.803$ | $0.782 \pm 0.028$ | $0.750$ |

For `hyper-batch ens` we usually start with a log-uniform distribution over the hyperparameters $p_t$ over the full range for the given bounds of the hyperparameters. For the ResNet models we find that reducing the initial ranges of $p_t$ for the $L_2$ regularization parameters by one order of magnitude is more stable (but we keep the original bounds for clipping the parameters).

**Tuning of optimization method hyperparameters.** We perform an exhaustive ablation of the different algorithmic choices for `hyper-batch ens` as well as for `batch ens` using the validation set. We run a grid search procedure evaluating up to five different values for each parameter listed below and repeat each run three times using different seeds. We find that the following configuration works best.

Shared parameters for both methods:

Figure 6: CIFAR-10. Comparison of our hyper-deep ensemble with deep ensemble, for different ensemble sizes in terms of cross entropy (negative log-likelihood), accuracy, Brier score and expected calibration error for a Wide ResNet 28-10 over CIFAR-10.

- The base learning rate of SGD: $0.1$.
- Learning rate decay ratio: $0.2$.
- Batch size: $64$.
- Initialization of each entry of the fast weights according to $\mathcal{N}(1,1)$.
- We multiply the learning rate for the fast weights by: $2.0$.

Parameters specific to hyper-batch ensemble:

- Range for the $L_2$ parameters: $[0.1, 100]$.
- Range for the label smoothing parameter: $[0, 0.2]$.
- Entropy regularization parameter: $\tau = 10^{-3}$ (as also used in the other experiments and used by [52]).
- Learning rate for the tuning step (where we use Adam): $10^{-5}$.

Remarkably, we find that the shared set of parameters which work best for batch ensembles, also work best for hyper-batch ensembles. This makes our method an easy-to-tune drop-in replacement for batch ensembles.

### D.2 Regularization of the rank-1 factors

As explained in the previous section, we find that for the Wide ResNet architecture, both hyper-batch ensemble and batch ensemble work best when the rank-1 factors ($\mathbf{r}_k \mathbf{s}_k^\top$ and $\mathbf{u}_k \mathbf{v}_k^\top$) are not regularized. We examine the performance of both models when *using* a regularization of the rank-1 factors. For these versions of the models, we run an ablation over the same algorithmic choices as done in the previous section. The results are displayed in Table 10. The performance of both methods is substantially worse than the unregularized versions as presented in the main text, Table 2.

### D.3 Out-of-distribution experiments

In this section, we provide an out-of-distribution evaluation along the line of Table 1 in [32]. More precisely, for each of the four approaches `deep ens`, `hyper-deep ens`, `batch ens` and `hyper-batch ens`, we compute on out-of-distribution samples from other image datasets the following metrics:

- Mean maximum confidence (MMC) on out-distribution samples (lower is better)

Figure 7: CIFAR-10. Additional plots for calibration on CIFAR-10 corruptions. The boxplots show a comparison of expected calibration error and cross entropy (negative log-likelihood) on different levels of corruption. Each box shows the quartiles summarizing the results across all types of skew while the error bars indicate the min and max across different skew types. The plot for accuracy under corruptions can be found in Figure 3.

Table 11: Out-of-distribution evaluation based on other image datasets. The table reports MMC (↓)/ AUROC (↑) / FPR@95 (↓) (see the precise definitions of the metrics in the text).

| | Trained on CIFAR-100 | | Trained on CIFAR-10 | |
| | CIFAR-10 | SVHN | CIFAR-100 | SVHN |
| --- | --- | --- | --- | --- |
| deep ens (4) | 0.502 / 0.816 / 0.762 | 0.538 / 0.796 / 0.792 | 0.742 / 0.912 / 0.482 | 0.599 / 0.972 / 0.185 |
| hyper-deep ens (4) | 0.524 / 0.822 / 0.741 | 0.580 / 0.787 / 0.787 | 0.730 / 0.915 / 0.469 | 0.608 / 0.967 / 0.237 |
| batch ens (4) | 0.568 / 0.810 / 0.753 | 0.594 / 0.795 / 0.771 | 0.800 / 0.908 / 0.493 | 0.700 / 0.961 / 0.269 |
| hyper-batch ens (4) | 0.544 / 0.814 / 0.748 | 0.553 / 0.813 / 0.753 | 0.746 / 0.907 / 0.519 | 0.675 / 0.951 / 0.364 |

- The AUC of the ROC curve (AUROC) for the task of discriminating between in- and out-distributions based on the confidence value (higher is better)

- The false positive rate at 95% true positive rate (FPR@95) in the same discriminative task (lower is better).

We summarize the results in Table 11, where we consider models both trained on CIFAR-10 (with evaluation on CIFAR-100 and SVHN) and CIFAR-100 (with evaluation on CIFAR-10 and SVHN). In a nutshell, `hyper-deep ens` (respectively `hyper-batch ens`) tends to favourably compare with `deep ens` (respectively `batch ens`) on CIFAR-10 and CIFAR-100, while they appear to perform worse over SVHN.

### D.4 Complementary results for CIFAR-10

In this section we show complementary results to those presented in the main text for CIFAR-10. Figure 6 compares hyper-deep ensembles against deep ensembles for varying ensemble sizes. We find that the performance gain on CIFAR-10 is not as substantial as on CIFAR-100 presented in Figure 1. However, `hyper-deep ens` improves upon `deep ens` for large ensemble sizes in terms of NLL (cross entropy) and expected calibration error (ECE). The accuracy of `hyper-deep ens` is slightly higher for most ensemble sizes (except for ensemble sizes 3 and 10).

Figure 7 shows a comparison of additional metrics on the out of distribution experiment presented in the main text, Figure 3. We observe the same trend as in Figure 3 that `hyper-batch ens` is more robust than `batch ens` as it typically leads to smaller worst values (see top whiskers in the boxplot).

### D.5 Complementary results for CIFAR-100

In this section we show complementary results to those presented in the main text for CIFAR-100. Figure 8 presents additional metrics (Brier score and expected calibration error) for varying ensemble sizes for hyper-deep ensemble and deep ensemble. Additionally to the strong improvements in terms of accuracy and NLL presented in Figure 1, we find that `hyper-deep ens` also improves in terms of Brier score and but is slightly less calibrated than deep ensemble for large ensemble sizes.

Figure 8: CIFAR-100. Comparison of our hyper-deep ensemble with deep ensemble, for different ensemble sizes, in terms of Brier score and expected calibration error for a Wide ResNet 28-10 over CIFAR-100. Plots for negative log-likelihood and accuracy can be found in the main text, in Figure 1.

Table 12: Comparison of the numbers of parameters and training cost for `hyper-batch ens` and `batch ens` for Wide ResNet 28-10.

| CIFAR-10 | Time/epoch. | total epochs. | total time | # parameters |
|---|---|---|---|---|
| `hyper-batch ens` | 2.07 min. | 300 | 10.4h | 73.1M |
| `batch ens` | 1.01 min. | 250 | 4.2h | 36.6M |
| CIFAR-100 | | | | |
| `hyper-batch ens` | 2.16 min. | 300 | 10.8h | 73.2M |
| `batch ens` | 1.10 min. | 250 | 4.6h | 36.6M |

### D.6 Memory and training time cost

For hyper-batch ensemble and batch ensemble, Table 12 reports the training time and memory cost in terms of number of parameters. Our method is roughly twice as costly as batch ensemble with respect to those two aspects. As demonstrated in the main text, this comes with the advantage of achieving better prediction performance. In Appendix C.7.4 we show that doubling the number of parameters for batch ensemble still leads to worse performance than our method.

## E   Towards more compact self-tuning layers

The goal of this section is to motivate the introduction of different, more compact parametrizations of the layers in self-tuning networks.

In [52], the choice of their parametrization (i.e., shifting and rescaling) is motivated by the example of ridge regression whose solution is viewed as a particular 2-layer linear network (see details in Section B.2 of [52]). The parametrization is however not justified for other losses beyond the square loss. Moreover, by construction, this parametrization leads to at least a 2x memory increase compared to using the corresponding standard layer.

If we take the example of the dense layer with input and output dimensions $r$ and $s$ respectively, recall that we have

$$\mathbf{W} + \mathbf{\Delta} \circ \mathbf{e}(\boldsymbol{\lambda}), \text{ with } \mathbf{W}, \mathbf{\Delta} \in \mathbb{R}^{r \times s}.$$

Let us denote by $\boldsymbol{\zeta}_j \in \{0, 1\}^s$ the one-hot vector with its $j$-th entry equal to 1 and 0 elsewhere, $e_j(\boldsymbol{\lambda})$ the $j$-th entry of $\mathbf{e}(\boldsymbol{\lambda})$ and $\boldsymbol{\delta}_j$ the $j$-th column of $\mathbf{\Delta}$. We can rewrite the above equation as

$$\mathbf{W} + \sum_{j=1}^{s} e_j(\boldsymbol{\lambda})\boldsymbol{\delta}_j\boldsymbol{\zeta}_j^\top = \mathbf{W} + \sum_{j=1}^{s} e_j(\boldsymbol{\lambda})\mathbf{W}_j = \sum_{j=0}^{s} e_j(\boldsymbol{\lambda})\mathbf{W}_j \text{ with } e_0(\boldsymbol{\lambda}) = 1 \text{ and } \mathbf{W}_0 = \mathbf{W}. \quad (14)$$

As a result, we can re-interpret the parametrization of [52] as a very specific linear combination of parameters $\mathbf{W}_j$ where the coefficients of the combination, i.e., $\mathbf{e}(\boldsymbol{\lambda})$, depend on $\boldsymbol{\lambda}$.

Based on this observation and insight, we want to further motivate the use of self-tuned layers with more general linear combinations (dependent on $\boldsymbol{\lambda}$) of parameters, paving the way for more *compact* parametrizations. For instance, with $\mathbf{W} \in \mathbb{R}^{r \times s}, \mathbf{G} \in \mathbb{R}^{r \times h}$ and $\mathbf{H} \in \mathbb{R}^{s \times h}$ as well as $\mathbf{e}(\boldsymbol{\lambda}) \in \mathbb{R}^h$,

we could consider

$$\mathbf{W} + \sum_{j=1}^{h} e_j(\boldsymbol{\lambda})\mathbf{g}_j\mathbf{h}_j^{\top} = \mathbf{W} + (\mathbf{G} \circ \mathbf{e}(\boldsymbol{\lambda}))\mathbf{H}^{\top}. \tag{15}$$

Formulation (15) comes with two benefits. On the one hand, it reduces the memory footprint, as controlled by the low-rank factor $h$ which impacts the size of both $(\mathbf{G} \circ \mathbf{e}(\boldsymbol{\lambda}))\mathbf{H}^{\top}$ and $\mathbf{e}(\boldsymbol{\lambda})$. On the other hand, we can hope to get more expressiveness and flexibility since in (14), only the $\boldsymbol{\delta}_j$'s are learned, while in (15), both the vector $\mathbf{g}_j$'s and $\mathbf{h}_j$'s are learned.

### E.1 Problem statement

Along the line of [52], but with a broader scope, beyond the ridge regression setting, we now provide theoretical arguments to justify the use of such a parametrization. We focus on the linear case with arbitrary convex loss functions. We start by recalling some notation, some of which slightly differ from the rest of the paper.

**Notations.** In the following derivations, we will use

- Input point $\mathbf{x} \in \mathbb{R}^d$ with target $y$
- The distribution over pair $(\mathbf{x}, y)$ is denoted by $\mathcal{P}$
- Domain $\Lambda \subseteq \mathbb{R}^{m+1}$ of $(m+1)$-dimensional hyperparameter $\boldsymbol{\lambda} = (\lambda_0, \boldsymbol{\lambda}_1) \in \Lambda$ (with $\boldsymbol{\lambda}_1$ of dimension $m$). We split the vector representation to make explicitly appear $\lambda_0$, the regularization parameter, for a reason that will be clear afterwards.
- Feature transformation of the input points $\phi : \mathbb{R}^d \mapsto \mathbb{R}^k$. When the feature transformation is itself parametrized by some hyperparameters $\boldsymbol{\lambda}_1$, we write $\phi_{\boldsymbol{\lambda}_1}(\mathbf{x})$.
- The distribution over hyperparameters $(\lambda_0, \boldsymbol{\lambda}_1)$ is denoted by $\mathcal{Q}$
- Embedding of the hyperparameters $\mathbf{e} : \Lambda \mapsto \mathbb{R}^q$
- The loss function $\hat{y} \mapsto \ell_{\boldsymbol{\lambda}_1}(y, \hat{y})$, potentially parameterized by some hyperparameters $\boldsymbol{\lambda}_1$.

We focus on the following formulation

$$\min_{\mathbf{U} \in \mathbb{R}^{k \times q}} \mathbb{E}_{(\lambda_0, \boldsymbol{\lambda}_1) \sim \mathcal{Q}} \left[ \mathbb{E}_{(\mathbf{x},y) \sim \mathcal{P}} \left[ \ell_{\boldsymbol{\lambda}_1}(y, \phi_{\boldsymbol{\lambda}_1}(\mathbf{x})^{\top} \mathbf{U}\mathbf{e}(\boldsymbol{\lambda})) \right] + \frac{\lambda_0}{2} \|\mathbf{U}\mathbf{e}(\boldsymbol{\lambda})\|^2 \right]. \tag{16}$$

Note the generality of (16) where the hyperparameters sampled from $\mathcal{Q}$ influence the regularization term (via $\lambda_0$), the loss (via $\ell_{\boldsymbol{\lambda}_1}$) and the data representation (with $\phi_{\boldsymbol{\lambda}_1}$).

In a nutshell, we want to show that, for any $\boldsymbol{\lambda} \in \Lambda$, $\mathbf{U}\mathbf{e}(\boldsymbol{\lambda})$—i.e., a linear combination of parameters whose combination depends on $\boldsymbol{\lambda}$, as in (15)—can well approximate the solution $\mathbf{w}(\boldsymbol{\lambda})$ of

$$\min_{\mathbf{w} \in \mathbb{R}^k} \mathbb{E}_{(\mathbf{x},y) \sim \mathcal{P}} \left[ \ell_{\boldsymbol{\lambda}_1}(y, \phi_{\boldsymbol{\lambda}_1}(\mathbf{x})^{\top} \mathbf{w}) \right] + \frac{\lambda_0}{2} \|\mathbf{w}\|^2.$$

In Proposition E.3, we show that when we apply a stochastic optimization algorithm to (16), e.g., SGD or variants thereof, with solution $\hat{\mathbf{U}}$, it holds in expectation over $\boldsymbol{\lambda} \sim \mathcal{Q}$ that $\mathbf{w}(\boldsymbol{\lambda}) \approx \hat{\mathbf{U}}\mathbf{e}(\boldsymbol{\lambda})$ under some appropriate assumptions.

Our analysis operates with a fixed feature transformation $\phi_{\boldsymbol{\lambda}_1}$ (e.g., a pre-trained network) and with a fixed embedding of the hyperparameters $\mathbf{e}$ (e.g., a polynomial expansion). In practice, those two quantities would however be learnt simultaneously during training. We stress that, despite those two technical limitations, the proposed analysis is more general than that of [52], in terms of both the loss functions and the hyperparameters covered (in [52], only the squared loss and $\lambda_0$ are considered).

We define (remembering the definition $\boldsymbol{\lambda} = (\lambda_0, \boldsymbol{\lambda}_1)$)

$$\begin{aligned} g_{\boldsymbol{\lambda}}(\mathbf{w}) &= \mathbb{E}_{(\mathbf{x},y) \sim \mathcal{P}} \left[ \ell_{\boldsymbol{\lambda}_1}(y, \phi_{\boldsymbol{\lambda}_1}(\mathbf{x})^{\top} \mathbf{w}) \right] \\ f_{\boldsymbol{\lambda}}(\mathbf{U}\mathbf{e}(\boldsymbol{\lambda})) &= g_{\boldsymbol{\lambda}}(\mathbf{U}\mathbf{e}(\boldsymbol{\lambda})) + \frac{\lambda_0}{2} \|\mathbf{U}\mathbf{e}(\boldsymbol{\lambda})\|^2 \\ F(\mathbf{U}) &= \mathbb{E}_{\boldsymbol{\lambda} \sim \mathcal{Q}} \left[ f_{\boldsymbol{\lambda}}(\mathbf{U}\mathbf{e}(\boldsymbol{\lambda})) \right] = \mathbb{E}_{\boldsymbol{\lambda} \sim \mathcal{Q}} \left[ g_{\boldsymbol{\lambda}}(\mathbf{U}\mathbf{e}(\boldsymbol{\lambda})) \right] + \frac{1}{2} \mathrm{Tr}(\mathbf{U}\mathbf{C}\mathbf{U}^{\top}) \end{aligned}$$

## E.2 Assumptions

**(A1)** For all $\boldsymbol{\lambda} \in \Lambda$, $g_{\boldsymbol{\lambda}}(\cdot)$ is convex and has $L_{\boldsymbol{\lambda}}$-Lipschitz continuous gradients.

**(A2)** The matrices $\mathbf{C} = \mathbb{E}_{\boldsymbol{\lambda} \sim \mathcal{Q}}[\lambda_0 \mathbf{e}(\boldsymbol{\lambda})\mathbf{e}(\boldsymbol{\lambda})^{\top}]$ and $\boldsymbol{\Sigma} = \mathbb{E}_{\boldsymbol{\lambda} \sim \mathcal{Q}}[\mathbf{e}(\boldsymbol{\lambda})\mathbf{e}(\boldsymbol{\lambda})^{\top}]$ are positive definite.

## E.3 Direct consequences

Under the assumptions above, we have the following properties:

- For all $\boldsymbol{\lambda} \in \Lambda$, the problem

$$\min_{\mathbf{w} \in \mathbb{R}^k} \left\{ g_{\boldsymbol{\lambda}}(\mathbf{w}) + \frac{\lambda_0}{2} \|\mathbf{w}\|^2 \right\}$$

  admits a unique solution which we denote by $\mathbf{w}(\boldsymbol{\lambda})$. Moreover, it holds that

$$\nabla g_{\boldsymbol{\lambda}}(\mathbf{w}(\boldsymbol{\lambda})) + \lambda_0 \mathbf{w}(\boldsymbol{\lambda}) = \mathbf{0} \tag{17}$$

- $F(\cdot)$ is strongly convex ($\mathbf{C} \succ \mathbf{0}$) and the problem

$$\min_{\mathbf{U} \in \mathbb{R}^{k \times q}} F(\mathbf{U})$$

  admits a unique solution which we denote by $\mathbf{U}^{\star}$.

## E.4 Preliminary lemmas

Before listing some lemmas, we define for any $\boldsymbol{\lambda} \in \Lambda$ and any $\boldsymbol{\Delta}(\boldsymbol{\lambda}) \in \mathbb{R}^k$

$$R(\boldsymbol{\Delta}(\boldsymbol{\lambda})) = g_{\boldsymbol{\lambda}}(\boldsymbol{\Delta}(\boldsymbol{\lambda}) + \mathbf{w}(\boldsymbol{\lambda})) - g_{\boldsymbol{\lambda}}(\mathbf{w}(\boldsymbol{\lambda})) - \boldsymbol{\Delta}(\boldsymbol{\lambda})^{\top} \nabla g_{\boldsymbol{\lambda}}(\mathbf{w}(\boldsymbol{\lambda}))$$

which is the residual of the first-order Taylor expansion of $g_{\boldsymbol{\lambda}}(\cdot)$ at $\mathbf{w}(\boldsymbol{\lambda})$. Given Assumption **(A1)**, it notably holds that

$$0 \le R(\Delta(\boldsymbol{\lambda})) \le \frac{L_{\boldsymbol{\lambda}}}{2} \|\boldsymbol{\Delta}(\boldsymbol{\lambda})\|_2^2. \tag{18}$$

**Lemma E.1.** *We have for any $\mathbf{U} \in \mathbb{R}^{k \times q}$ and any $\boldsymbol{\lambda} \in \Lambda$, with $\Delta(\boldsymbol{\lambda}) = \mathbf{U}\mathbf{e}(\boldsymbol{\lambda}) - \mathbf{w}(\boldsymbol{\lambda})$,*

$$
\begin{aligned}
f_{\boldsymbol{\lambda}}(\mathbf{U}\mathbf{e}(\boldsymbol{\lambda})) &= g_{\boldsymbol{\lambda}}(\mathbf{U}\mathbf{e}(\boldsymbol{\lambda})) + \frac{\lambda_0}{2}\|\mathbf{U}\mathbf{e}(\boldsymbol{\lambda})\|^2 \\
&= g_{\boldsymbol{\lambda}}(\boldsymbol{\Delta}(\boldsymbol{\lambda}) + \mathbf{w}(\boldsymbol{\lambda})) + \frac{\lambda_0}{2}\|\boldsymbol{\Delta}(\boldsymbol{\lambda}) + \mathbf{w}(\boldsymbol{\lambda})\|^2 \\
&= f_{\boldsymbol{\lambda}}(\mathbf{w}(\boldsymbol{\lambda})) + R(\boldsymbol{\Delta}(\boldsymbol{\lambda})) + \frac{\lambda_0}{2}\|\boldsymbol{\Delta}(\boldsymbol{\lambda})\|^2 + \boldsymbol{\Delta}(\boldsymbol{\lambda})^{\top}[\nabla g_{\boldsymbol{\lambda}}(\mathbf{w}(\boldsymbol{\lambda})) + \lambda_0 \mathbf{w}(\boldsymbol{\lambda})] \\
&= f_{\boldsymbol{\lambda}}(\mathbf{w}(\boldsymbol{\lambda})) + R(\boldsymbol{\Delta}(\boldsymbol{\lambda})) + \frac{\lambda_0}{2}\|\boldsymbol{\Delta}(\boldsymbol{\lambda})\|^2
\end{aligned}
$$

*where in the last line we have used the optimality condition (17) of $\mathbf{w}(\boldsymbol{\lambda})$.*

As a direct consequence, we have the following result:

**Lemma E.2.** *For any $\mathbf{U}_1, \mathbf{U}_2 \in \mathbb{R}^{k \times q}$ and defining for any $\boldsymbol{\lambda} \in \Lambda$, with $\boldsymbol{\Delta}_j(\boldsymbol{\lambda}) = \mathbf{U}_j \mathbf{e}(\boldsymbol{\lambda}) - \mathbf{w}(\boldsymbol{\lambda})$, it holds that*

$$F(\mathbf{U}_1) \le F(\mathbf{U}_2)$$

*if and only if*

$$\mathbb{E}_{\boldsymbol{\lambda} \sim \mathcal{Q}}\left[R(\boldsymbol{\Delta}_1(\boldsymbol{\lambda})) + \frac{\lambda_0}{2}\|\boldsymbol{\Delta}_1(\boldsymbol{\lambda})\|^2\right] \le \mathbb{E}_{\boldsymbol{\lambda} \sim \mathcal{Q}}\left[R(\boldsymbol{\Delta}_2(\boldsymbol{\lambda})) + \frac{\lambda_0}{2}\|\boldsymbol{\Delta}_2(\boldsymbol{\lambda})\|^2\right].$$

## E.5 Main proposition

Before presenting the main result, we introduce a key quantity that will drive the quality of our guarantee. To measure how well we can approximate the family of solutions $\{\mathbf{w}(\boldsymbol{\lambda})\}_{\boldsymbol{\lambda} \in \Lambda}$ via the choice of $\mathbf{e}$ and $\mathcal{Q}$, we define

$$\mathbf{U}_{\text{app}} = \arg\min_{\mathbf{U} \in \mathbb{R}^{k \times q}} \mathbb{E}_{\boldsymbol{\lambda} \sim \mathcal{Q}}\left[\|\mathbf{U}\mathbf{e}(\boldsymbol{\lambda}) - \mathbf{w}(\boldsymbol{\lambda})\|_2\right] \quad \text{and} \quad \boldsymbol{\Delta}_{\text{app}}(\boldsymbol{\lambda}) = \mathbf{U}_{\text{app}}\mathbf{e}(\boldsymbol{\lambda}) - \mathbf{w}(\boldsymbol{\lambda}).$$

The definition is unique since according to **(A2)**, we have $\boldsymbol{\Sigma} \succ \mathbf{0}$.

**Proposition E.3.** *Let assume we have an, possibly stochastic, algorithm $\mathcal{A}$ such that after $t$ steps of $\mathcal{A}$ to optimize ([16](#)), we obtain $\mathbf{U}_t$ satisfying*

$$\mathbb{E}_{\mathcal{A}}[F(\mathbf{U}_t)] \leq F(\mathbf{U}^\star) + \varepsilon_t^{\mathcal{A}}$$

*for some tolerance $\varepsilon_t^{\mathcal{A}} \geq 0$ depending on both $t$ and the algorithm $\mathcal{A}$. Denoting by $\mathbf{\Delta}_t(\boldsymbol{\lambda}) = \mathbf{U}_t \mathbf{e}(\boldsymbol{\lambda}) - \mathbf{w}(\boldsymbol{\lambda})$ the gap between the estimated and actual solution $\mathbf{w}(\boldsymbol{\lambda})$ for any $\boldsymbol{\lambda} \in \Lambda$, it holds that*

$$\mathbb{E}_{\mathcal{A},\,\boldsymbol{\lambda}\sim\mathcal{Q}}\Big[R(\mathbf{\Delta}_t(\boldsymbol{\lambda})) + \frac{\lambda_0}{2}\|\mathbf{\Delta}_t(\boldsymbol{\lambda})\|^2\Big] \leq \mathbb{E}_{\boldsymbol{\lambda}\sim\mathcal{Q}}\Big[R(\mathbf{\Delta}_{\text{app}}(\boldsymbol{\lambda})) + \frac{\lambda_0}{2}\|\mathbf{\Delta}_{\text{app}}(\boldsymbol{\lambda})\|^2\Big] + \varepsilon_t^{\mathcal{A}}.$$

*In particular, we have:*

$$\mathbb{E}_{\mathcal{A},\,\boldsymbol{\lambda}\sim\mathcal{Q}}\Big[\lambda_0\|\mathbf{U}_t\mathbf{e}(\boldsymbol{\lambda}) - \mathbf{w}(\boldsymbol{\lambda})\|^2\Big] \leq \mathbb{E}_{\boldsymbol{\lambda}\sim\mathcal{Q}}\Big[(L_{\boldsymbol{\lambda}} + \lambda_0)\|\mathbf{\Delta}_{\text{app}}(\boldsymbol{\lambda})\|^2\Big] + \varepsilon_t^{\mathcal{A}}.$$

*Proof.* Starting from

$$\mathbb{E}_{\mathcal{A}}[F(\mathbf{U}_t)] \leq F(\mathbf{U}^\star) + \varepsilon_t^{\mathcal{A}}$$

and applying Lemma [E.2](#), we end up with (the expectation $\mathbb{E}_{\mathcal{A}}$ does not impact the result of Lemma [E.2](#) since the term $f_{\boldsymbol{\lambda}}(\mathbf{w}(\boldsymbol{\lambda}))$ that cancels out on both sides is not affected by $\mathcal{A}$)

$$\mathbb{E}_{\mathcal{A},\,\boldsymbol{\lambda}\sim\mathcal{Q}}\Big[R(\mathbf{\Delta}_t(\boldsymbol{\lambda})) + \frac{\lambda_0}{2}\|\mathbf{\Delta}_t(\boldsymbol{\lambda})\|^2\Big] \leq \mathbb{E}_{\boldsymbol{\lambda}\sim\mathcal{Q}}\Big[R(\mathbf{\Delta}^\star(\boldsymbol{\lambda})) + \frac{\lambda_0}{2}\|\mathbf{\Delta}^\star(\boldsymbol{\lambda})\|^2\Big] + \varepsilon_t^{\mathcal{A}}.$$

Similarly, by definition of $\mathbf{U}^\star$ as the minimum of $F(\cdot)$, we have

$$F(\mathbf{U}^\star) \leq F(\mathbf{U}_{\text{app}})$$

which leads to

$$\mathbb{E}_{\boldsymbol{\lambda}\sim\mathcal{Q}}\Big[R(\mathbf{\Delta}^\star(\boldsymbol{\lambda})) + \frac{\lambda_0}{2}\|\mathbf{\Delta}^\star(\boldsymbol{\lambda})\|^2\Big] \leq \mathbb{E}_{\boldsymbol{\lambda}\sim\mathcal{Q}}\Big[R(\mathbf{\Delta}_{\text{app}}(\boldsymbol{\lambda})) + \frac{\lambda_0}{2}\|\mathbf{\Delta}_{\text{app}}(\boldsymbol{\lambda})\|^2\Big].$$

Chaining the two inequalities leads to the first conclusion. The second conclusion stems from the application of ([18](#)). $\square$

## Footnotes

[2]The precise relationship between $\nu_{k,i}$ and $\boldsymbol{\lambda}_{k,i}$ depends on the implementation details and on how the hyperparameters of the problem, e.g., the dropout rates or $L_2$ penalties, are stored in the vector $\boldsymbol{\lambda}_{k,i}$.

[3]https://github.com/google/uncertainty-baselines/tree/master/baselines/cifar