[Reviews · NeurIPS 2020]

Review 1

Summary and Contributions: ++ Post Rebuttal I'm happy with the rebuttal which clarified some points about the paper. The extra experiments show that the method extends beyond WideResNet too. For this I'm raising my score. ++ This paper discusses the idea of combining ensemble through hyper-parameters and different initialization of a deep model. The paper applied this idea on two directions. The first one is on deep ensembles in which they introduced stratified hyper ensemble as a greedy search algorithm that updates over hyper-ensemble method. The second direction is in applying the idea on batch ensembles, which is a budget wise ensemble mechanism. They introduced batch hyper ensemble which 2x size of hyper ensemble. Also, it merged the idea of self-tuning networks into batch-hyper-ensemble to make an efficient upgrade-non greedy approach for batch-ensemble.

Strengths: I have read the paper several times(4-5) to make sure of these points: - I'm not sure if the idea is novel itself in this area or not but it seems valid. - The empirical evaluation shows an improvement over previous methods and it covered the comparison with different alternatives such as deep ensembles and batch ensembles. - The analysis of the pictorial view in figure 2 where they showed that deep ensembles and hyper ensembles parameters search are a special case of the method. - The upgrade of self-tuning networks to match K ensembles such as in equation 7 and the updated objective function in equation 8 is another contribution of the paper.

Weaknesses: I do have several questions: - Did the hyper-ensemble paper forced the networks to start form the same initialization point? I looked into the paper in ref[12] for this information but couldn't tell. If so, then the work will need a more justification with the difference w.r.t hyper-ensemble. - In page 5, starting from line 172 it was not clear why o(mk) became o(k^2), can you elaborate more on this? - From table#1 and table#2 it seems that hyper-ens, str hyper ens and deep ens are quite close to each other in nll,acc,ece ranges. What's exactly the range if improvement of using str-hyper-ens over the others? - In tables#1,2 what is the meaning of the numbers in brackets (1),(4)? - I understand that the empirical evaluation is expensive, but reporting results on other deep models such as VGG, ResNet, DenseNet for a small subset of the settings will clear any doubts regards that the method only works best for wide-resnet. - On the same point of wide-resnet as in lines 269-272 for using two deep ensembles, what are the results of this comparison for wideresnet?

Correctness: It seems correct expect for the comments in the weaknesses section.

Clarity: Each section of the paper is clear on its own but the overall flow is not the best. Because the paper have a main idea [hyper parameters + different initialization] then it applied to two different techniques [deep ensemble and batch ensemble]. It's better to introduce the idea first then a complete section for the improvement over deep ensemble and same for batch ensemble. Same for the discussion in the empirical evaluation section. Of course this is a suggestion.

Relation to Prior Work: Yes.

Reproducibility: Yes

Additional Feedback:


Review 2

Summary and Contributions: Post-rebuttal: ----------------------------------------------- Thank you for the clarification. After reading the other reviews and the rebuttal, I decided to increase my score. ----------------------------------------------- This paper presents a generalization of deep ensembles (Lakshminarayanan et al., NIPS 2017). In addition to ensembling neural networks based on distinct points in the parameter space, this paper proposes to also the *hyper*parameter space (which contains all possible values of neural networks hyperparameters, e.g. weight decay, dropout rate, learning rate). The authors propose a greedy algorithm for constructing an ensemble given a set of models which have different parameter and hyperparameter values. Furthermore, a lightweight version of this algorithm, based on the recently proposed batch ensembles (Wen et al., ICLR 2020), is proposed. Extensive experiments show that ensembling networks based on both parameters and hyperparameters yield a substantial improvement in uncertainty quantification compared to the baselines.

Strengths: I like this paper since it is an important step toward full uncertainty quantification (i.e. quantifying all sources of uncertainty) of neural networks. As more sources of uncertainty are quantified, the quality of predictive uncertainty---the quantity that matters the most in predictive systems---improves. This can have a big implication in safety-critical systems since one can trust the predictive uncertainty better. Empirical evaluation is solid but can be further improved: It is sufficiently broad but it could be more in-depth. Please find some suggestions in the bottom of this review.

Weaknesses: This paper has a strong connection to the hierarchical Bayesian modeling of neural networks, where a hyperprior (prior distribution over hyperparameters) is assigned to the probabilistic model. Hyper ensembles can roughly be seen as consisting of samples from the posterior of this Bayesian model. It is thus a bit disappointing that the authors did not compare, discuss, or at least mention this connection in the paper. A comparison and discussion would be very helpful to point out exactly the novelty of hyper ensembles compared to this established Bayesian modeling technique.

Correctness: The proposed methods are straightforward generalizations of prior works such as deep & batch ensembles and self-tuning networks (MacKay et al., ICLR 2019), so I do not think there is any obvious issue here. Nevertheless, there is a questionable design decision in the method: In lines 157-159: Why does the algorithm hyper_ens select a model *with replacement*? Doesn't this mean that the resulting ensemble could consist of K exact copies of a single model? In this case, wouldn't it defeat the purpose of forming ensembles?

Clarity: This paper is very well written. I appreciate the extensive discussion about deep & batch ensembles and self-tuning networks. One minor complaint would be: what does "skew" mean in the caption of Figure 3? I do not think that it is discussed or defined anywhere in the main text.

Relation to Prior Work: I think the authors have discussed the related work sufficiently well, except the connection to hierarchical Bayesian models. I would like to see this connection to be discussed and possibly compared in the experiment section. Additionally, I think (standard, non-hierarchical) BNN baselines need to be compared in the empirical evaluation.

Reproducibility: Yes

Additional Feedback: Additional feedback: - Please do not wait until acceptance before releasing the code. I think even a simple self-contained code example over a toy dataset in the form of a Jupyter notebook would be really helpful. - I would like to see more OOD experiment. Perhaps a big table consisting of AUROC values like Table 1 in Hein et al., CVPR 2019? A deeper OOD experiment is important since it will complement the frequentist calibration results in Table 1 and 2 (frequentist calibration only concerns about in-distribution uncertainty). - Please polish the References section. There are some inaccuracies there, e.g. [25] and [28]---they are either workshop or conference papers, not just Arxiv papers. Questions: - What is the interpretation of the parameter \xi_t in the distribution of p(\lambda | \xi_t)? Does this mean p(\lambda | \xi_t) is some kind of (time-dependent) stochastic process? - How many samples do you use for computing the objectives in Eq. 8 and 9? To summarize, I think this paper provides an important step toward full uncertainty quantification---where all sources of uncertainty are considered---of neural networks. I think this paper will be much stronger if (i) the authors discussed the connection between hyper ensemble and hierarchical Bayes, (ii) added more non-ensemble baselines like BNNs, and (ii) added deeper OOD experiments.


Review 3

Summary and Contributions: 1. This paper unifies hyper-parameter tuning and random initialization as two dimensions to encourage model diversity. When combining these two methods, the overall result is better than each method. 2. The paper further applies a recently proposed batch ensemble technique to simulate deep ensemble and extend the existing self-tuning networks to the ensemble learning scenario. 3. Empirical results are provided on benchmark datasets with different architectures.

Strengths: Empirical results look believable and the authors promise to release code upon acceptance.

Weaknesses: 1. The proposed method marginally improve over previous methods. 2. The proposed method is a combination of existing techniques. The main innovation I can see so far is the design of self-tuning networks for ensemble learning. 3. The paper claims that two sources of diversity jointly contribute to the overall ensemble model. Actually, there is a third source of diversity during training from p_t(\lambda_k) that controls the diversity of \lambda_k. Assuming p_t will not degenerate, how to effectively control the variance of p_t such that it can do a good local search job around \lambda_k? It would be great if there is a qualitative explanation that multiple ensemble members p_t(\lambda_k) for k=1,..., K work well independently and can jointly explore a wider space of lambda.

Correctness: All techniques are properly used in this paper as far as I can see.

Clarity: This paper is well-written.

Relation to Prior Work: This paper includes related prior works as far as I can see.

Reproducibility: Yes

Additional Feedback:


Review 4

Summary and Contributions: This paper proposed to do ensembles over both weights and hyperparameters to improve the performance. Specifically, the paper proposed stratified hyper ensembles that involves a random search over different hyperparameters and stratified across multiple random initializations. The authors also proposed batch hyper ensembles, which is a parameter efficient version of the model. The proposed model is tested on image classification tasks and achieves favorable performance.

Strengths: 1. This paper is well-motivated and well-written. It is easy to read and follow, which sufficient details on the model. 2. The performance of the model outperformance the baselines consistently.

Weaknesses: On the novelty. The proposed is simple and straightforward. Although the empirical performance is good, the novelty is incremental.

Correctness: Correct

Clarity: Yes

Relation to Prior Work: Yes

Reproducibility: Yes

Additional Feedback:

[Author Response · NeurIPS 2020]

We thank the four reviewers for their insightful comments and suggestions. Below, we have addressed most of the items given the time- and space-bounded aspects of the rebuttal, hoping we clarified the main questions of the reviewers.

**Reviewer 1 (R1):**

• "...I looked into the paper in ref[12] ...": In [12], the greedy algorithm is generic, with no assumptions about models it forms an ensemble from. In particular, the models are not forced to start from the same initialization, which we will clarify in the paper. For hyper ensemble, we are further interested in using a fixed initialization to isolate the effect of just varying the hyperparameters (while deep ensembles vary only the initialization, with fixed hyperparameters).

• "...why o($mk$) became o($k^2$) ...": Random search leads to a set of $m$ models. If we were to stratify all of them, we would need $k$ seeds for each of those $m$ models, hence a total of O($mk$) models to train. However, if we first apply the greedy procedure to extract $k$ models out of the $m$ available ones, then the stratification needs $k$ seeds for each of those $k$ models, thus O($k^2$) models to train (as a reminder, the greedy procedure does not imply any training).

• "...hyper-ens, str hyper ens and deep ens are quite close to each other ...": Recent work like [16] show that improvements on WRN-cifar10/100 benchmarks are typically in small ranges (with larger room for improvements on cifar100). For Tab. 1, we ran the Wilcoxon signed-rank test (paired along settings, datasets and model types) and observe statistically significant improvements (except for ece, known to be noisier Nixon et al. (2020)). Similar results were obtained with a paired t-test. For Tab. 2 (with more costly experiments), we do not have enough runs to apply such tests. We nonetheless report the standard errors in the paper, which seem to indicate significant improvements.

|  | ens size | p-value (nll) | p-value (acc) | p-value (ece) | ens size | p-value (nll) | p-value (acc) | p-value (ece) |
|---|---|---|---|---|---|---|---|---|
| deep ens ↔ str hyper ens | 3 | $1.1{\times}10^{-5}$ | $2.1{\times}10^{-5}$ | 0.25 | 5 | $9.1{\times}10^{-6}$ | $1.9{\times}10^{-5}$ | 0.33 |
| hyper ens ↔ str hyper ens | 3 | 0.0725 | 0.0017 | 0.43 | 5 | 0.0088 | 0.0018 | 0.44 |

• "...numbers in brackets ...": Those numbers indicate the size of the ensemble; we will clarify this point.

• "...reporting results on other deep models ...": We thank R1 for the idea and ran our entire benchmark for ResNet-20:

| ResNet-20 / cifar100 | nll ($\downarrow$) | acc ($\uparrow$) | ece ($\downarrow$) | ResNet-20 / cifar100 | nll ($\downarrow$) | acc ($\uparrow$) | ece ($\downarrow$) |
|---|---|---|---|---|---|---|---|
| single (1) | 1.245 | 0.679 | 0.105 | batch ens (4) | 1.235 | 0.697 | 0.119 |
| deep ens (4) | 0.905 | 0.749 | 0.043 | batch hyper ens (4) | 1.141 | 0.702 | 0.059 |
| str hyper ens (4) | 0.905 | 0.751 | 0.048 | | | | |

**Reviewer 2 (R2):**

• "...hierarchical Bayesian modeling of neural networks ...": Hyper ensembles can indeed be viewed as a mixture variational posterior and the entropy penalty is the ELBO's KL divergence toward a uniform prior. There are many related works from Bayes, e.g., Kemp & Tenenbaum (2008), Adams et al. (2009), Grosse et al. (2012), Lake et al. (2015). They typically use Bayes nonparametric priors/posteriors and MCMC; we use mixtures and SGD. We will add more detailed discussion to the paper.

• "...with replacement ...": When used *with replacement*, the greedy algorithm from Caruana et al. [12, Sec. 2.1] makes it possible to find a *weighted* combination of models (e.g., $\frac{1}{4}$ (2 model$_a$ + model$_b$ +model$_c$) would correspond to the situation where model$_a$ has been selected twice). To avoid the pitfall rightly mentioned by R2, Algorithm 1 and Algorithm 2 (in appendix) make use of ".unique()" to correctly count the number of members.

• "...skew ...": Skew intensity refers to the severity of the distortion applied to the corrupted dataset; see [28, 55].

• "...BNN baselines ...": We use the same data/training/evaluation pipeline as that used in the `baselines` of the `edward2` repository. We can thus directly compare with the reported metrics for BNN VI and MC dropout on cifar10/100. E.g., on cifar100: nll/acc/ece=0.944/0.778/0.097 and 0.830/0796/0.050, which we will add in the paper.

• "...OOD experiment ...": We thank R2 for this suggestion. Along the line of Tab. 1 in Hein et al. (CVPR 2019), we computed the table below for our WRN experiments (MMC/AUROC/FPR@95 are defined in Hein et al. (2019))

|  | (trained on cifar100, MMC ($\downarrow$)/ AUROC ($\uparrow$) / FPR@95 ($\downarrow$)) | | (trained on cifar10, MMC ($\downarrow$)/ AUROC ($\uparrow$) / FPR@95 ($\downarrow$)) | |
|---|---|---|---|---|
|  | cifar10 | SVHN | cifar100 | SVHN |
| deep ens | 0.502 / 0.818 / 0.758 | 0.495 / 0.826 / 0.756 | 0.737 / 0.914 / 0.477 | 0.644 / 0.964 / 0.265 |
| str hyper ens | 0.525 / 0.823 / 0.744 | 0.561 / 0.802 / 0.764 | 0.727 / 0.917 / 0.455 | 0.572 / 0.973 / 0.172 |
| batch ens | 0.626 / 0.810 / 0.784 | 0.621 / 0.825 / 0.796 | 0.806 / 0.907 / 0.504 | 0.681 / 0.968 / 0.211 |
| batch hyper ens | 0.583 / 0.811 / 0.748 | 0.574 / 0.823 / 0.736 | 0.714 / 0.911 / 0.507 | 0.634 / 0.956 / 0.329 |

• "...interpretation of the parameter $\xi_t$ ...": In our setting, the parameter $\xi_t$ contains the lower and upper bounds of the log-uniform distribution at the step $t$. Given $\xi_t$, $p(\lambda|\xi_t)$ is a standard log-uniform distribution.

• "...How many samples do you use for computing the objectives in Eq. 8 and 9? ...": We use one sample for each data point in the batch. Sec. 5.1 (MLP and LeNet) uses 256. Sec. 5.2 (WRN) uses 512—64 for each of 8 workers.

**Reviewer 3 (R3):**

• "...third source of diversity ...": The distribution $p_t$ is log-uniform. Its variance is implicitly controlled by the entropy regularization (see Eq. 9) since both the variance and entropy depend on the width of the support (see lines 232 to 234). At prediction time, the variance does not play an explicit role since we use the mean of $p_t$, like in [45] (see lines 230-231). However the variance has a direct impact during the optimization when the $\lambda_k$'s are sampled.

• "...qualitative explanation ...": While the $K$ distributions $p_t(\lambda_k)$'s are independent, we stress that their parameters $\{\xi_{k,t}\}_{k=1}^{K}$ are *jointly* learned in the tuning phase (see Eq. 9). Indeed, the *ensemble cross-entropy loss* ties together the $K$ members (and hence $\{p(\lambda_k|\xi_{k,t})\}_{k=1}^{K}$). E.g., we see the complementarity of the members by comparing the ensemble metrics (nll/acc=0.718/0.821; see Tab. 2 in the paper) with the *average ensemble-member* metrics (nll/acc=0.851/0.804).

**Reviewer 4 (R4):** We thank R4 for the comments and feedback.

[Meta-Review · NeurIPS 2020]

All 4 reviewers push for accepting this paper. The excellent coverage of the state-of-the-art and the quality of experiments make up for a more limited novelty. A release of code will be essential for the significance of this work which could become a State-of-the-Art reference.